# Single-photon microscopy to study biomolecular condensates

Eleonora Perego [1,6], Sabrina Zappone [1,2,6], Francesco Castagnetti[3], Davide Mariani [3], Erika Vitiello[3], Jakob Rupert[4,5], Elsa Zacco [4], Gian Gaetano Tartaglia [4,5], Irene Bozzoni[3,5], Eli Slenders [1] & Giuseppe Vicidomini [1] ✉

Biomolecular condensates serve as membrane-less compartments within cells, concentrating proteins and nucleic acids to facilitate precise spatial and temporal orchestration of various biological processes. The diversity of these processes and the substantial variability in condensate characteristics present a formidable challenge for quantifying their molecular dynamics, surpassing the capabilities of conventional microscopy. Here, we show that our single-photon microscope provides a comprehensive live-cell spectroscopy and imaging framework for investigating biomolecular condensation. Leveraging a single-photon detector array, single-photon microscopy enhances the potential of quantitative confocal microscopy by providing access to fluorescence signals at the single-photon level. Our platform incorporates photon spatio-temporal tagging, which allowed us to perform time-lapse super-resolved imaging for molecular sub-diffraction environment organization with simultaneous monitoring of molecular mobility, interactions, and nano-environment properties through fluorescence lifetime fluctuation spectroscopy. This integrated correlative study reveals the dynamics and interactions of RNA-binding proteins involved in forming stress granules, a specific type of biomolecular condensates, across a wide range of spatial and temporal scales. Our versatile framework opens up avenues for exploring a broad spectrum of biomolecular processes beyond the formation of membrane-less organelles.

Fluorescence fluctuation spectroscopy (FFS) is a family of techniques that allows quantifying molecular concentration, mobility, interaction, confinement, and aggregation in crowded and complex environments such as living cells[1–3]. These methods capture the variation in fluorescence intensity resulting from fluorescently labeled molecules passing through a small detection volume, typically on the order of femtoliters. The analysis of these fluctuations is essential to characterize the properties of molecules. Among the FFS methods, fluorescence correlation spectroscopy (FCS) stands out as the most commonly employed. In FCS, the autocorrelation function of the intensity fluctuations is analyzed in order to grant averaged information about molecular mobility and concentration of the sample.

In laser-scanning microscopy (LSM), FCS relies on confocality or two-photon excitation to generate the small detection volume (or probed region) and employs a fast single-point detector, such as a photomultiplier (PMT) or a single-photon avalanche diode (SPAD), to sample the fluorescence signal. However, recent advancements have demonstrated that the information content of LSM-based FCS

[1]Molecular Microscopy and Spectroscopy, Istituto Italiano di Tecnologia, Genoa, Italy. [2]Dipartimento di Informatica, Bioingegneria, Robotica e Ingegneria dei Sistemi, University of Genoa, Genoa, Italy. [3]Non coding RNAs in Physiology and Pathology, Istituto Italiano di Tecnologia, Genoa, Italy. [4]RNA Systems Biology, Istituto Italiano di Tecnologia, Genoa, Italy. [5]Department of Biology and Biotechnologies 'C. Darwin', Sapienza University of Rome, Rome, Italy. [6]These authors contributed equally: Eleonora Perego, Sabrina Zappone. ✉e-mail: giuseppe.vicidomini@iit.it

increases by replacing the conventional single-point detector with a fast (≤μs) array detector capable of directly image the probed region[4–6]. Unlike single-point detectors, which average out the spatial variations in the fluorescence signal from the probed region, fast array detectors preserve spatial information without compromising temporal sampling. This spatial information has been harnessed to perform several FFS methods with a single measurement and in a simpler configuration, named comprehensive correlation analysis[4], thereby providing more insights into the studied diffusive processes. One such method, termed spot-variation FCS (svFCS), infers sub-diffraction limited spatial heterogeneity within the probed region by measuring the diffusion times in tunable observation volume sizes, all while still capturing molecular mobility[7,8]. As a result, svFCS allows for the discrimination between free or constrained motion induced by sub-diffraction structures, such as trapping within partition domains or hopping through meshwork[9].

In conventional svFCS, the tunable detection volumes are obtained through a series of sequential measurements, involving either physical modification of the optical setup, such as altering the back-aperture objective lens, or the application of super-resolution techniques, such as stimulated emission depletion microscopy[10–12]. However, these approaches come at the cost of reduced versatility and increased photo-toxicity. An alternative approach involves the creation of different detection volumes using array detectors, where the signal from selected array elements is integrated in a post-processing phase[13,14]. With this method, it becomes possible to extract transit times as a function of multiple effective detection volumes from a single measurement, significantly mitigating phototoxicity. Moreover, svFCS based on an array detector can uncover temporal heterogeneity in motion types on the scale of a single measurement, typically lasting tens of seconds. This capability addresses a limitation of conventional svFCS, where such information is typically lost.

The first implementation of comprehensive correlation analysis used a PMT array, specifically the AiryScan detector[15]. However, these implementation introduces complexity in data calibrations and analysis due to the PMT analog read-out. The solution to this problem came with the introduction of asynchronous read-out SPAD array detectors[16,17], which not only resolved the calibration and analysis issues but also brought about new perspectives. When integrated with dedicated data acquisition (DAQ) systems, asynchronous read-out SPAD array detectors enable photon-resolved measurement of the fluorescence signal from the detection volume: every photon is tagged with a comprehensive set of spatial and temporal signatures. By directing the output of each SPAD element to a multi-channel time-tagging DAQ module (see Fig. 1a), we can assign a spatial tag to every single photon, corresponding to its arrival position within the detector plane, projected into the sample plane, with sub-diffraction precision. Additionally, each photon is assigned a time-tag corresponding to the delay (tens of picoseconds precision) from the experiment start (i.e., absolute time). Specifically, each photon is tagged with the spatial position of the probed region, i.e., where is located the detection volume with respect to the sample, and its delay from the fluorophore excitation event. This value is typically termed start-stop time or arrival time. The additional photon spatial tag allows for the reconstruction of a (raster) image or the implementation of scanning FCS.

Here, we present a comprehensive framework of live-cell spectroscopy and imaging experiments that leverages this massive single-photon information dataset to investigate biomolecular condensates in living cells. We show the potential of our platform to study complex biological processes, such as biomolecular condensation, which spans several spatial-temporal scales. The membraneless condensates are composed of proteins, typically with intrinsically disordered domains, and nucleic acids, e.g., DNA and RNA. They are governed by weak and multivalent interactions between the molecules and tend to have spherical shapes, fluidity, and liquid-like properties[18,19]. The membraneless condensates are involved in different cell cycle stages, helping maintain cellular stability. It is believed that molecules in the intracellular space can form reaction compartments by phase-separating in

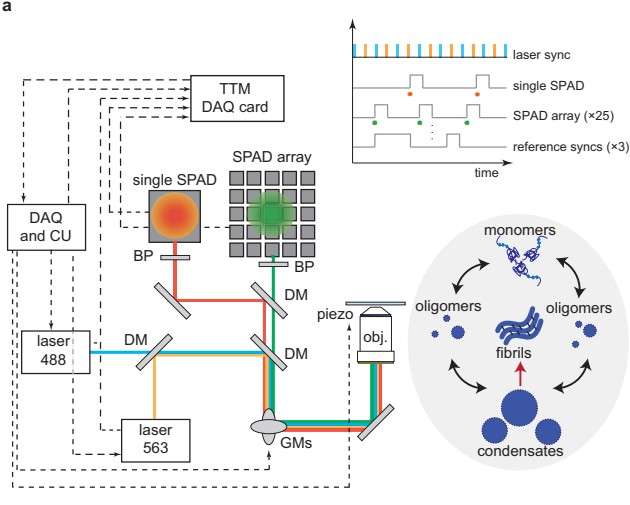

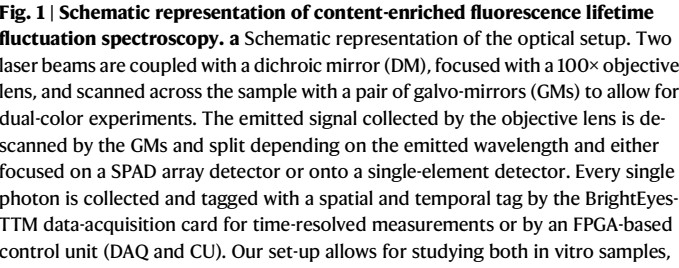

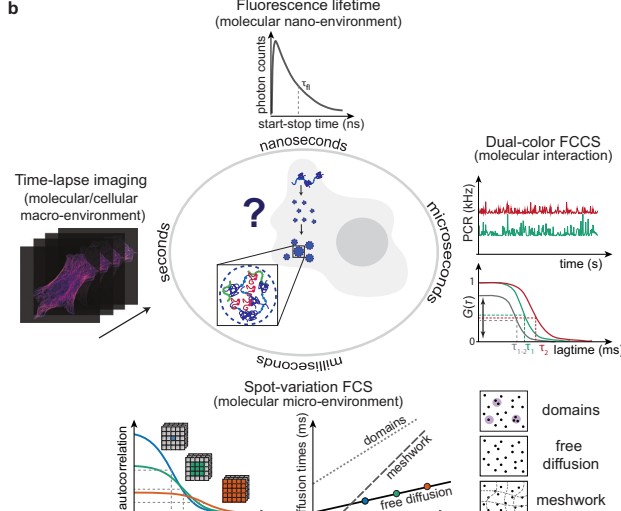

**Fig. 1 | Schematic representation of content-enriched fluorescence lifetime fluctuation spectroscopy. a** Schematic representation of the optical setup. Two laser beams are coupled with a dichroic mirror (DM), focused with a 100× objective lens, and scanned across the sample with a pair of galvo-mirrors (GMs) to allow for dual-color experiments. The emitted signal collected by the objective lens is de-scanned by the GMs and split depending on the emitted wavelength and either focused on a SPAD array detector or onto a single-element detector. Every single photon is collected and tagged with a spatial and temporal tag by the BrightEyes-TTM data-acquisition card for time-resolved measurements or by an FPGA-based control unit (DAQ and CU). Our set-up allows for studying both in vitro samples, such as purified proteins, and living cells. **b** Thanks to the spatiotemporal tags of the SPAD array detector, we can access several temporal scales. The fluorescence lifetime, on the nanosecond time regime, which gives us information about the nano-environment of single molecules, is measured simultaneously with the microseconds intensity fluorescence fluctuation, giving us information about the diffusion. Granted by the single-photon spatial tags, the diffusion times in the different detected areas can be evaluated (diffusion law), providing information about diffusion modality or the micro-environment of the molecules. The temporal and spatial tags can be provided in a multi-color scheme, allowing us to perform dual-color cross-correlation measurements.

condensates and creating an environment where multiple biological reactions co-occur[20].

We focused on the formation of stress granules (SGs), which are composed mostly by mRNAs and RNA-binding proteins and their formation triggers cellular stress response[21]. Their persistent formation or aberration, such as the inclusion of mutated proteins, has been related to neurodegenerative diseases[22]. In particular, some mutations of FUS (FUsed in Sarcoma) protein are related to amyotrophic lateral sclerosis (ALS). FUS is a pleiotropic RNA binding protein with several functions in regulating RNA metabolism; although being mainly nuclear, ALS-causing mutations trigger its displacement inside the cytoplasm. This results in a double toxic effect, as it combines a loss of function in the nucleus and a gain of toxic function in the cytoplasm. In its ALS-mutated forms, FUS is also incorporated into the SGs upon stress induction, dysregulating consequently SGs physiology[23]. It still remains unclear whether the mislocation and the co-localization of the ALS-mutated FUS inside SGs are causative of ALS or whether this cytopathological aggregation is the response consequence to the stress of degenerated motoneurons. The aggregation of mutated FUS could inhibit the localization of RNAs encoding for proliferating factors, hindering cell recovery after stress and thus increasing neuronal loss. Moreover, the RNA-binding capability of FUS might increase the sequestration of specific mRNAs, which might be essential for neuronal viability.

Due to the complexity of the process, only with a multi-parameter platform and on multiple spatiotemporal scales, it is possible to get a complete view of the processes behind SG formation and eventually on SG aberration and aggregation. Combining the SPAD array detector with the BrightEyes-TTM, our multi-channel time-tagging DAQ module, grants us access to sample information across multiple spatial and temporal scales (Fig. 1b). The single-photon arrival time, which requires access to the variations of the fluorescence signal at the nanosecond time scale, enables calculating the fluorescence lifetime of the labeled biomolecule (Fig. 1b top). Simultaneously, the single-photon absolute time can be binned on the microseconds time scale, creating a time trace of fluorescence fluctuation for each element of the SPAD array detector. The intensity time traces can be analyzed in several ways to extract different physical quantities in the same experiment. Here, we focused on two approaches based on FFS: svFCS, to simultaneously monitor the molecular mobility and its sub-diffraction environment organization, and dual-color fluorescence cross-correlation spectroscopy (dcFCCS) to monitor interactions between two molecular species[24]. By labeling with two spectrally separable dyes two different molecules, the fluctuations of the fluorescence intensities can be acquired simultaneously for both species, and the two signals correlated with each other (cross-correlation spectroscopy). The relative amplitude of the cross-correlation curve is proportional to the molecular interaction level in the sample[24]. A very low amplitude, or a zero cross-correlation, is related to two molecules moving independently, while a high amplitude is caused by a coupled movement, indicating an interaction (Fig. 1b right). Since the SPAD array detector is integrated into a LSM, the single-photon absolute time and its spatial position within the sample allows for time-lapse super-resolved image scanning microscopy (ISM)[25,26] (Fig. 1b left). Notably, both imaging and spectroscopy methods are correlated with the fluorescence lifetime information. While our research group has previously demonstrated many of these methods individually, including svFCS, ISM, and fluorescence lifetime spectroscopy, what was lacking was an integrated platform that brought all these techniques together.

Our platform provides multiple parameters that are simultaneously quantified and correlated (i.e., nano-environment, sub-diffraction environment, mobility, interaction, and macro-environment) across different spatial and temporal scales for high-informative fluorescence experiments. While our focus has been on this particular biological process, it is important to note that this framework can be universally applied to various other processes involving the intricate dynamics of biomolecules.

## Results

### Validation of fluorescence lifetime fluctuation spectroscopy

To validate our multi-parameter platform, we started by using test samples. We first measured yellow-green (YG) fluorescent beads dissolved in water. We implemented svFCS with a 5 × 5 elements SPAD array detector and a custom laser-scanning microscope as explained in[5,27]. We used the photon absolute time to obtain the intensity time trace for each SPAD array element. We integrated the intensity traces to virtually detect the signal from three observation volumes: the volume from the central element only, the sum of the inner square (Sum 3 × 3), and the sum of all channels (Sum 5 × 5). We auto-correlated the resulting time traces and fitted the relative auto-correlation functions with a one-component 3D diffusing model (Fig. 2a). The diffusion law $\tau_D(\omega_0^2)$ allows distinguishing different modalities of diffusion[5,8,28]. In fact, the linear regression of the diffusion times at different observation volumes is described by the function $\tau_D = \omega_0^2/4D + t_0$[8,29]. The slope is related to the diffusion coefficient, while the intercept $t_0$ intuitively describes the time deviation compared to the expected time if the molecule's movement were solely governed by Brownian motion. This deviation is influenced by the sample environment. In the case of pure Brownian motion, $t_0 = 0$. However, for hopping diffusion, $t_0$ is greater than zero, and for diffusion through a meshwork, $t_0$ is less than zero.

Here, as the intercept is close to the origin $0.03 \pm 0.03$ ms, the diffusion law suggests an expected free diffusing motion (Fig. 2b). The measured diffusion coefficient $17.7 \pm 0.3 \ \mu m^2 \, s^{-1}$ and the calculated diameter ($d = 24$ nm) correspond well with previous measurements[5,6,27] and the manufacturer's information. We further validate our svFCS approach on a more complex system where mobility is not only restricted to free diffusion. We investigated the in vitro liquid-liquid phase separation (LLPS) of alpha-synuclein protein ($\alpha$-syn). Alpha-synuclein is a pre-synaptic protein expressed typically in the neuronal brain tissue. Solid-like aggregates of $\alpha$-syn have been identified as one of the main components of Lewy's bodies, the pathological neuronal inclusions associated with different neuronal diseases such as Parkinson's[30]. Recently, it has been shown that the phase separation in liquid-like condensates could precede the aggregation of soluble $\alpha$-syn[31]. Upon the induction of the LLPS in a crowded environment (mimicked by the molecular crowding poly-ethylene glycol), we followed the condensation process by imaging and svFCS (Supplementary Fig. 1).

We characterize the dynamics of monomeric $\alpha$-syn before the LLPS process. Imaging only shows a uniform solution of fluorescent $\alpha$-syn. Likewise, svFCS reveals one diffusing component. As expected, $\alpha$-syn is freely moving in solution with a diffusion coefficient of $D = 130 \pm 7 \ \mu m^2 \, s^{-1}$, comparable with previously reported values for monomeric $\alpha$-syn[32]. Once $\alpha$-syn proteins start the phase-separation process, round droplet-like condensates are visible by imaging. In this case, svFCS reveals two diffusing components in the solution. One, which corresponds to the dilute phase, and one, which corresponds to condensates. Interestingly, we can use the diffusion law $\tau(\omega_0^2)$ to distinguish the two different types of movement discovered by FCS fitting. The fast component shows a free-diffusion type of motion (intercept $t_0$ close to 0), suggesting that this component is mainly the dilute phase. The slower component shows a positive intercept, indicating a domain-confined diffusion. We disrupted the weak hydrophobic interactions responsible for condensate formation by adding the alcohol 1,6-hexanediol[33,34]. As expected, upon adding 10% of 1,6-hexanediol a change in the autocorrelation curve can be appreciated. Even if two diffusing components are still detected, the slower diffusing component shows a lower confinement inside the condensates

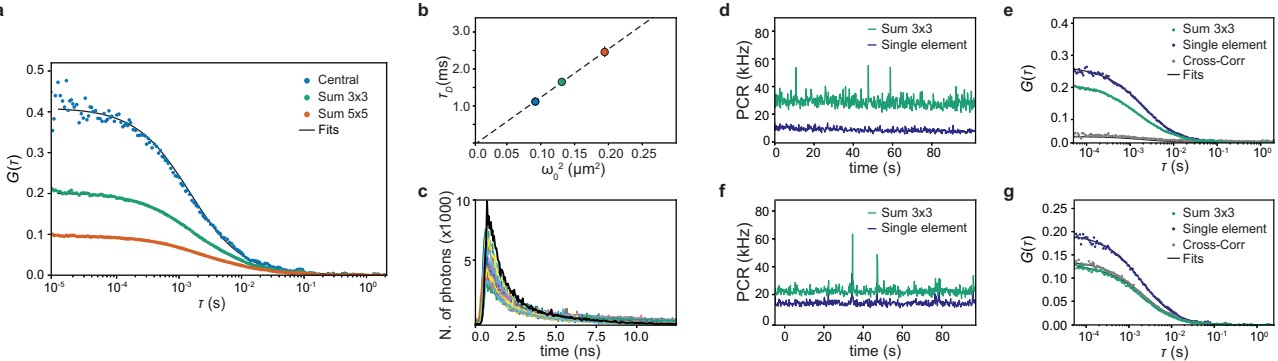

**Fig. 2 | Single measurement of fluorescent nanospheres with the SPAD array detector. a** FCS measurement of fluorescent nanospheres in water. The three curves correspond to the three volumes considered with the SPAD array detector. The three autocorrelation curves are fitted with a model for one diffusing component (black lines). **b** Spot-variation FCS measurement of fluorescent nanospheres in water. The diffusing times are calculated from the autocorrelation curves shown in (**a**). Here, the diffusion law confirms a free motion with an intercept value of $-0.03 \pm 0.03$ ms and a diffusion coefficient of $17.7 \pm 0.3\ \mu m^2\ s^{-1}$, which corresponds to a diameter of 24 nm (20 nm from the manufacturing). **c** Histogram of the fluorescence decay calculated from the same dataset used in (**a**). A histogram of the fluorescence decay is calculated for each of the 25 channels of the SPAD array detector. The black line represents the data from the central detector channel. The fluorescence lifetime value can be retrieved by fitting the data with an exponential

function ($\tau_F = 1.3 \pm 0.1$ ns). **d**, **e** Cross-correlation experiment with fluorescent nanospheres. A mix of beads with two colors is used for the measurements (yellow-green and red). The SPAD array detector is equipped with a filter for green fluorescence signals, while the single-element detector has a filter for red fluorescence signals. While the signal is autocorrelated in every detector alone (only the Sum $3 \times 3$ is shown here for the SPAD array detector), the cross-correlation shows a low amplitude, suggesting low interaction between the two samples. **f**, **g** Only yellow-green fluorescent nanospheres are diluted in water and measured. The signal is split equally on the two detector arms equipped with the same detection filter. The intensity time trace can be well-correlated in both detectors (only the Sum $3 \times 3$ is shown here for the SPAD array detector). As expected here, the cross-correlation curve has a high amplitude as the signal is the same for both detectors. Source data are provided as a Source Data file and raw files.

as well as a lower diffusion coefficient, indicating the starting point of the dissolution of the liquid-like condensates. The experiments on the LLPS of $\alpha$-syn confirm the effectiveness of svFCS on LLPS processes, being able to distinguish the changes in the sub-diffraction environment during condensation processes.

As the datasets are photon-resolved, we demonstrate how the same dataset can be evaluated with other methods, such as fluorescence lifetime. On the dataset of the YG beads in Fig. 2a, we calculated the photon-arrival time histograms for each SPAD array element (Fig. 2c). The photon-arrival time histogram measures the fluorescence decay distribution. The fluorescence lifetime of the tagged molecule is extracted by fitting the decay function. The fluorescence lifetime is known to be a reporter of the nano-environment of the fluorophores[35]. It has been shown how, by using lifetime-sensitive fluorescence probes or Förster resonance energy transfer (FRET) construct, the change in fluorescence lifetime is linked with changes in the biological nano-environment of single molecules[36]. As a proof-of-principle, we quantified the fluorescence lifetime value for the YG fluorescent beads, obtaining a value of $1.3 \pm 0.1$ ns.

Compared to our earlier studies[5,27], we have also upgraded our custom microscope to facilitate cross-correlation FCS. This enhancement allows us to delve deeper into exploring potential molecular interactions. No change in the data acquisition system and only minimal changes in the optical architecture are required. We have incorporated a single-point SPAD detector into the detection path, seamlessly integrated with the BrightEyes-TTM platform. This setup enables simultaneous signal recording from both the single-point SPAD and the SPAD array detector. Notably, our current BrightEyes-TTM can accommodate up to 50 channels. The fluorescence signal from the detection volume is split between the two detectors using either a 50-50 beam-splitter or a dichroic filter. The latter allows for spectrally separating the fluorescent signal. We tested the system by measuring a mixture of YG and red fluorescent beads. The single-element detector has been equipped with detection filters for the red emission, while the SPAD array detector for the green emission (Fig. 1a). We can spectrally resolve the fluorescence intensity and separate the signal on the two detectors (Fig. 2d). The autocorrelation functions for each color give us a measurement of the diffusion

coefficients for each species of beads (Supplementary Fig. 2). However, the cross-correlation function has a low amplitude, suggesting a shallow interaction between the species (Fig. 2e). If we use the same microscope configuration and measure only red or YG beads, one of the two autocorrelation functions becomes zero. Specifically, the autocorrelation function from the SPAD detector is zero for red-only beads, while the autocorrelation function from the single-element detector is zero for YG-only beads (see Supplementary Fig. 2). As a result, the dual-color cross-correlation curves in both cases show low amplitudes. However, if we instead measure only yellow-green fluorescent beads using a 50-50 beam-splitter in the detection path and green detection filters also on the single-element detector, we can obtain both good autocorrelation curves and a cross-correlation curve with a high amplitude (Fig. 2f, g).

## Spot-variation FCS reveals protein condensation in living cells

In the context of life sciences, applying our platform allows the simultaneous investigation of molecular function and structure in combination with the temporal evolution of dynamic processes like aggregation, condensation, or interactions. We used our platform to investigate the formation of stress granules (SGs) in neuroblastoma-like (SK-N-BE) cells under oxidative stress. We focused mainly on Ras GTPase-activating protein-binding protein (G3BP1), a core component of SG assembly network[37]. We measured its interaction with FUsed in Sarcoma (FUS) protein in a WT cell model and in a pathological-related mutated model. In the latter, FUS protein has a single point mutation (proline instead of leucine) at the 525 position, which leads to a mis-placement of the protein from the nucleus to the cytoplasm, and it is related to amyotrophic lateral sclerosis (ALS) disease[38,39]. To follow FUS and G3BP1 dynamics, we took advantage of SK-N-BE stable cell lines expressing FUS protein, WT or P525L mutated, fused to RFP under a doxycycline-inducible promoter[40]. Both WT and mutant cell lines were also selected for the stable expression of G3BP1 protein tagged with eGFP under the constitutive Eif1a promoter, and the formation of SGs was triggered by applying 0.5 mM sodium arsenite to the cells.

First, we characterized the sample before inducing the oxidative stress. Figure 3a shows a SK-N-BE WT cell uniformly expressing G3BP1-

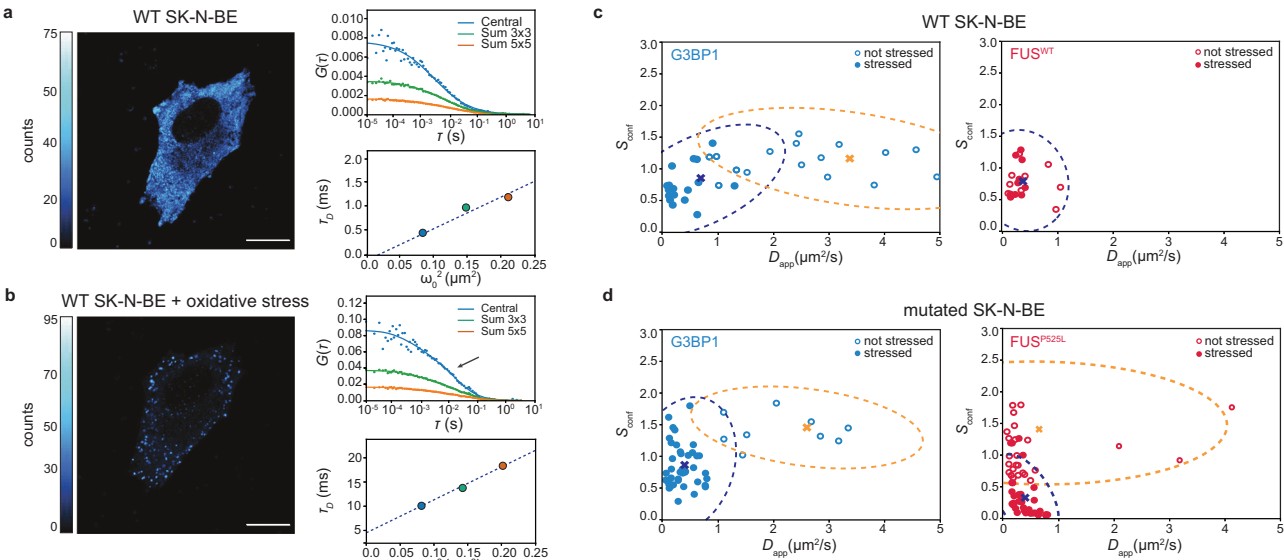

**Fig. 3 | Spot variation fluorescence spectroscopy for LLPS processes in living cells. a** ISM intensity-based image of an SK-N-BE cell WT expressing G3BP1-eGFP before oxidative stress. A svFCS measurement is performed in the cytoplasm by acquiring the fluorescence intensity over time and calculating, off-line, the autocorrelation curves for the three observation volumes (central, Sum 3 × 3 and Sum 5 × 5). The diffusing times can be plotted against the area of the observation volume to retrieve the modality of diffusion with the diffusion law. Scale bar 10 µm. **b** ISM intensity-based image of an SK-N-BE cell WT expressing G3BP1-eGFP during oxidative stress. SGs are visible in the cytoplasm. The change of diffusion/modality of diffusion is reflected in the autocorrelation curves, calculated as in (**a**). The diffusing times is plotted against the area of the observation volume to retrieve the modality of diffusion with the diffusion law. Scale bar 10 µm. **c** Confinement strength in relation to the diffusion coefficient of G3BP1 (cyan) and FUS (magenta) protein in WT SK-N-BE cells before the stress (empty circle, $n = 20$ cells measured over three independent experiments for G3BP1 and $n = 9$ cells measured over three

independent experiments for FUS) and after the stress (full circle, $n = 17$ cells measured over three independent experiments for G3BP1 and $n = 10$ cells measured over two independent experiments for FUS). The dotted ellipses represent the covariance confidence level of each cluster found with the K-mean clustering algorithm. The crosses represent the centroid position for each cluster. **d** Confinement strength in relation to the diffusion coefficient of G3BP1 (cyan) and FUS (magenta) protein in mutated SK-N-BE cells before the stress (empty circle, $n = 9$ cells measured over three independent experiments for G3BP1 and $n = 24$ cells measured over three independent experiments for FUS) and after the stress (full circle, $n = 20$ cells measured over three independent experiments for G3BP1 and $n = 32$ cells measured over three independent experiments for FUS). The dotted ellipses represent the covariance confidence level of each cluster found with the K-mean clustering algorithm. The crosses represent the centroid position for each cluster. Source data are provided as a Source Data file.

eGFP in the cytoplasm. We performed svFCS to probe its mobility (one example is in Fig. 3a top-right) in different cells and different cell positions. The autocorrelation curves could be fitted with a one-component 3D diffusing model for the three detection volumes of the SPAD array detector. The diffusion law plot $\tau_D(\omega_0^2)$ (bottom-right) suggests, in this case, a meshwork-confined type of diffusion. Once the oxidative stress is induced, SGs are visible in cells (Fig. 3b), and the correlation curves show a second slower diffusing component in the fits (Fig. 3b top-right). The diffusion modality changed compared to before (positive intercept) the formation of the stress granules, indicating a change in the micro-environment of G3BP1 toward compartmentalized movement.

When dealing with multiple cell conditions, interpreting the diffusion law analysis can become challenging, particularly when cells exhibit intrinsic variability during dynamic processes (see Supplementary Fig. 3). To facilitate comparisons between conditions, we introduce a metric known as confinement strength ($S_{conf}$). This metric is defined as the ratio between the diffusion coefficients calculated for the smallest focal area (central pixel only) and the biggest focal area (Sum 5 × 5). A similar metric was already used to represent how strong molecules are confined in domains while diffusing[41]. When $S_{conf}$ is 1, the diffusion coefficients are not dependent on the detection volumes, which is the case when molecules are freely diffusing. If $S_{conf}$ is bigger than 1, the molecules are moving in a dense meshwork, while when $S_{conf}$ is smaller than 1, the molecules are showing a confined-domain type of movement. The confinement strength allows us to parameterize the movement of the investigated molecules and compare the different cellular conditions. In the WT condition, G3BP1 restricts

its mobility during oxidative stress (Fig. 3c left). The apparent diffusion coefficients (in the x-axis, empty symbols: pre-stress, filled symbols: in SGs), i.e., the ones measured with the whole detector area (Sum 5 × 5), are smaller after the SGs are formed. In addition, after the stress, the confinement strength is lower than one, indicating a domain-confined behavior. Contrarily, FUS proteins, in the WT cell condition do not change behavior or mobility (Fig. 3c right).

In the pathologically mutated cellular condition (SK-N-BE FUS$^{P525L+}$), where FUS protein is misplaced in the cell cytoplasm, G3BP1 behaves similarly to the WT condition. The diffusion coefficients of G3BP1 decrease in the stress condition and its confinement strength, which moves toward the confinement-domain region. In opposition to the WT cell condition, FUS shows different types of mobility before and after the stress. This suggests a change in the interaction or in the type of FUS molecules' motility. While their diffusion coefficients do not seem to change in the stress granule formation process, the way that FUS proteins move is different.

In both WT and mutated SK-N-BE cell, clear differences of diffusion coefficients of G3BP1 before ($D_{WT} = 2.3 ± 1.2 \, \mu m^2 \, s^{-1}$, $D_{P525L+} = 2.1 ± 0.8 \, \mu m^2 \, s^{-1}$) and after the stress ($D_{WT} = 0.4 ± 0.3 \, \mu m^2 \, s^{-1}$, $D_{P525L+} = 0.3 ± 0.2 \, \mu m^2 \, s^{-1}$) are measured, indicating a strong reduction in the protein mobility during the formation of the SGs (Supplementary Fig. 4 left). The confinement strength of G3BP1 shows a more complicated behavior, reflecting the several roles that G3BP1 has inside the SGs (Supplementary Fig. 5 left). In the WT cell condition, before the stress, G3BP1 is diffusing freely in the cytoplasm ($S_{conf} = 1.1 ± 0.2$), while once the SGs are assembled, the confinement strength is lower than one ($S_{conf} = 0.6 ± 0.2$), indicating a confinement

behavior in the granules. In the mutated cell condition, G3BP1 seems to move initially in a meshwork ($S_{conf} = 1.4 \pm 0.2$), while, upon the formation of SGs its mobility is confined in domains ($S_{conf} = 0.8 \pm 0.3$), confirming again the formation of liquid condensates.

Regarding FUS protein in both WT and mutated cell condition, we did not observe a change in the diffusion coefficients before ($D_{WT} = 0.5 \pm 0.3 \, \mu m^2 \, s^{-1}$, $D_{P525L+} = 0.2 \pm 0.1 \, \mu m^2 \, s^{-1}$) and after ($D_{WT} = 0.30 \pm 0.08 \, \mu m^2 \, s^{-1}$, $D_{P525L+} = 0.4 \pm 0.2 \, \mu m^2 \, s^{-1}$) the formation of the SGs (Supplementary Fig. 4 right). In fact, FUS protein is already partially segregated into condensates before the induction of oxidative stress (Fig. 6a). However, the two cellular conditions differ in the type of mobility of FUS protein. In the WT cell condition, we do not observe differences in the confinement strength in not-stressed, $0.8 \pm 0.2$ and stressed conditions, $0.9 \pm 0.3$. It is important to notice that FUS in the WT cell condition is found only in the nuclei and is not involved in forming SGs in the cytoplasm. On the contrary, in the mutated cells with FUS in the cytoplasm, the confinement strength significantly decreases from $1.0 \pm 0.4$ before the stress to $0.3 \pm 0.3$ in SGs (Supplementary Fig. 5 right), indicating a confinement-type of mobility. The large standard deviations in the values of the confinement strength for FUS protein are related to the biological variability encountered inside SGs. FUS proteins interact with multiple partners inside SGs, from other RNA-binding proteins to RNA molecules.

We performed a blind k-mean clustering algorithm that partitions the dataset in $k$ clusters, minimizing each observation's distance to the nearest mean. We performed the elbow's method to retrieve the number of clusters $k$ for the analysis. In almost all cases, we found that 2 clusters were enough to describe our datasets. In all cases, the clustering algorithm can classify the data points into two groups, which correspond well to the pre-stress and post-stress conditions. To visually assess the clustering algorithm and the biological variability of the not-stressed/stressed conditions, we calculated the covariance confidence level based on Pearson's coefficient (dotted ellipses in Fig. 3c, d).

We followed the SG formation process by investigating the cells' recovery from the stress condition over time. We investigated the cellular and sub-cellular level changes during stress retrieval by ISM time-lapse imaging (Supplementary Fig. 6a). The SGs dissolve over the course of 5 h. This proves the reversibility of the induced stress, confirming that the investigated process behind the SGs formation is a liquid-liquid phase separation. In between the ISM imaging, we performed single-point spectroscopy measurements in multiple positions of the cells. In this way, we have a time-lapse measurement of the mobility of the single proteins. The apparent diffusion coefficient and the confinement strength can be correlated with the kinetics of the recovery process (Supplementary Fig. 7). In Supplementary Fig. 6b, the data points are color-coded with the recovery time. The clustering algorithm shows a similar clusterization for the stress-inducing process (Fig. 3c, d). We observe an increase in the value of the diffusion coefficient over time due to the dissolution of the SGs (Supplementary Fig. 7). It's worth noting that our capabilities extend beyond merely observing the dissolution of the SGs through imaging. We can also directly assess changes at the level of single proteins, achieving a level of detail that surpasses the limits of optical resolution.

**Fluorescence lifetime reveals protein confinement**

Thanks to the precise photon arrival time provided by the BrightEyes-TTM platform, we are able to construct the photon arrival-time (or decay) histogram of the fluorescence signal collected by the SPAD array detector. This capability enables us to conduct fluorescence lifetime measurements. In the context of imaging, we demonstrated the potentiality of the SPAD array detector combined with the BrightEyes-TTM platform, by implementing fluorescence lifetime image scanning microscopy (FLISM): firstly, we used the adaptive pixel-reassignment (APR) algorithm to reconstruct the high spatial

resolution and high signal-to-noise ratio image-scanning microscopy (ISM) bi-dimensional intensity image[25,42] (Fig. 4a bottom-left, G3BP1-eGFP expressing cells under oxidative stress) and the three-dimensional ISM time-resolved image, where the third dimension represents the photon arrival time. Secondly, we used the time-resolved ISM image to quantify the fluorescence lifetimes. This was achieved by fitting the photon arrival-time histograms for each pixel with a single-component exponential decay function (Fig. 4a, b top-right) and by phasor analysis (Fig. 4c, d). Monitoring the changes in the fluorescence lifetime is crucial as it can be related to structural or functional changes in the cellular structure. We found here a significant (t-test, $p$ value = 2e−6) decrease in the fluorescence lifetime of G3BP1-eGFP when inside SGs (Fig. 4f). The phasor plot analysis on the ISM reconstructed data confirms a difference in the fluorescence lifetime of G3BP1-eGFP inside or outside SGs. Before stress induction, the phasor plot reveals the presence of one-lifetime component, with a cloud of points centered on the universal circle (Supplementary Fig. 8). Once the SGs are formed, the phasor plot moves inside the universal circle, confirming the presence of multiple components: one short, corresponding to proteins in SGs, and one long, protein in cytoplasm.

We leveraged the phasor plot of stressed cells to segment the ISM intensity-based images and facilitate real-time tracking of changes in the protein environment. Using a straightforward centroid-based segmentation approach, based on the phasor plot, we split the intensity-based image into two images: one related to pixels with shorter lifetimes and one related to pixels with longer lifetime (Fig. 4d). As we found that the fluorescence lifetime is dependent on the condensation state of G3BP1 (inside or outside SGs), we achieved an automatic spatial segmentation of SGs within individual cells without any assumptions, e.g., intensity thresholds or fitting procedures.

We further investigate SG dynamics testing the influence of RNaseA. Because RNA molecules are known to be inside SGs, their disruption is expected to influence SGs. We performed both fluorescence lifetime-based imaging (Fig. 5a–d) and spectroscopy in several cells (Fig. 5i, j). Cells with RNaseA have been measured for 30 min as the stress-recovery mechanism could also affect dynamics at longer times. The spectroscopy measurements have been performed in zoomed regions in the center of single SGs (Fig. 5i, j). In the first 30 min, we did not find a difference in the confinement strengths or in the diffusion coefficients. The spectroscopy measurement points lie in the middle of SG clusters for both mutated or WT cells. While the microsecond dynamics did not change, we found an increase over time of fluorescence lifetime, which could be the first sign of SG disassembly on the molecular level. We further investigate the RNaseA treatment, checking for cellular macro-changes by qualitatively assessing the ISM images. We used the centroid-based segmentation approach for each image, relying on the phasor plot to automatically segment SGs on ISM images (Fig. 5b–e). For each segmented image, both before the RNaseA treatment (Fig. 5c) and after (Fig. 5f), we apply a particle analysis algorithm to quantify the shape and the area of the SGs. We found that the addition of RNaseA in stressed cells expressing G3BP1-eGFP leads to a reduction in the size of SGs (Fig. 5h), with a more dramatic reduction in WT cells, as also described in ref. 43. This suggests that the co-presence of FUS$^{P525L+}$ in the mutated cells is possibly related to aberrant SGs with a more stable protein-protein interaction. For the WT case, we also found a decrease in the circularity of the SGs with RNaseA. The addition of RNaseA induced a dissolution of SGs, which are now composed of smaller and more dynamic condensates unpacking from a big SG. For the mutated case, the circularity is lower compared to the WT case even before the addition of RNaseA, suggesting a completely different structure of the SGs. While G3BP1 dynamics inside the structure seem to be unaffected, the shape of the SGs is dependent on the presence of FUS$^{P525L+}$ (Supplementary Fig. 9), as it was already found for other aberrant proteins recruited in SGs[43,44].

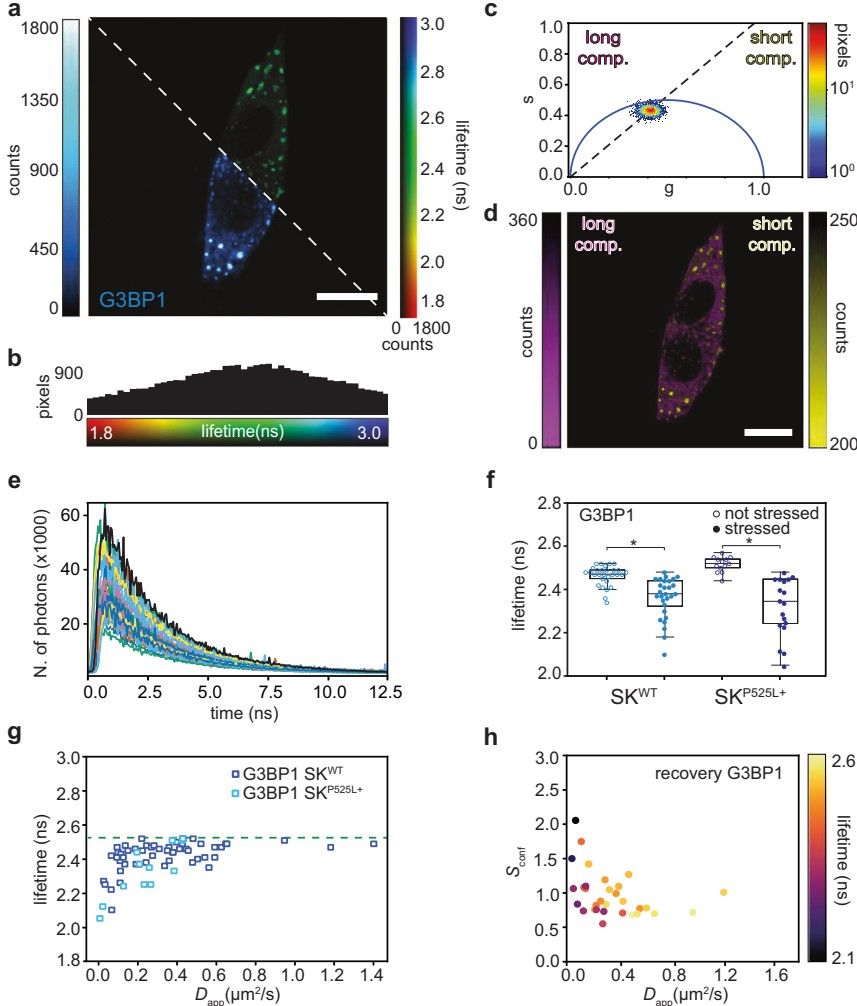

**Fig. 4 | Fluorescence lifetime fluctuation spectroscopy. a** SK-N-BE WT cell expressing G3BP1-eGFP under oxidative stress. Bottom-left ISM reconstructed intensity-based image, and top-right ISM reconstructed fluorescence lifetime-based image. Scale bar 10 μm. **b** Histogram of the fluorescence lifetime values of the image in (**a**). **c** Pixel intensity thresholded phasor plots. Number of pixels versus the polar coordinate (10% thresholds) of the image shown in (**a**). **d** Phasor-based segmentation. ISM intensity images were obtained by back projecting the points on the left of the phasor centroid (dotted line), representing pixels of the image with a long lifetime, and the points on the right of the phasor centroid, representing pixels of the image with a short lifetime. Scale bar 10 μm. **e** Time decay histograms for the different SPAD array detector pixels, central pixel in black. **f** Box-plots of G3BP1 fluorescence lifetime in SK-N-NE cells WT (left, $n = 29$ before stress, $n = 28$ during stress, two independent experiments) and mutated (right, $n = 13$ before stress,

$n = 18$ during stress, two independent experiments). Empty circles represent measurements in the cytosolic phase before the stress, and full circles represent measurements in the stress granule. In both cases, the fluorescence lifetimes are significantly different when measured in the diluted or in the condensed phase (Welch's $t$-test confidence level = 0.95, $p$ value = $3e^{-6}$ for WT and $p$ value = $1e^{-5}$ for mutated condition). The horizontal line in each box represents the median, the edges are the 25th and 75th percentile, and the vertical line extends to the minimum and maximum data points. **g** Fluorescence lifetime of G3BP1 in WT cells (dark blue, $n = 20$) and in the mutated cells (cyan, $n = 25$) correlated with the apparent diffusion coefficient. The dotted line corresponds to the fluorescence lifetime of free eGFP. **h** Fluorescence lifetime of G3BP1 in mutated cells correlated with the confinement strength and the diffusion coefficient during a recovery experiment ($n = 31$). Source data are provided as a Source Data file.

As a control, we tested if these behaviors were purely due to the cell internalization of the RNaseA by repeating the experiments on non-permeable SK-N-BE cells. In this case, no clear trend was measured in the SG area or in the circularity for both mutant and WT cells upon the addition of RNase (Supplementary Fig. 10).

In the context of FFS, we've demonstrated how we can enrich the information derived from imaging experiments through the implementation of fluorescence lifetime fluctuation spectroscopy. By extracting both photon absolute and arrival times from the same dataset, we have the capability to conduct both single-point fluorescence correlation spectroscopy (svFCS) and fluorescence lifetime measurements on the same experimental dataset. Specifically, we achieve this by temporally correlating the intensity across the three SPAD array detector volumes and simultaneously utilizing photon arrival times to construct time-decay histograms (see Fig. 4e). This

approach allows us to establish correlations between dynamic processes and sub-diffraction changes while also probing alterations in the nano-environment surrounding fluorescent-tagged molecules (see Fig. 4g). As a result, we gain valuable insights into the relationship between molecular structure and mobility. In our case, the fluorescence lifetime decreases in correlation with the diffusion coefficient for G3BP1 both in the WT and in the mutated cell conditions, indicating a rearrangement of the dye molecules during the SG formation. A decrease in fluorescence lifetime has already been reported during amyloid fibril formation[45]. The fluorescence lifetime reaches a plateau at a value around 2.55 ns, which corresponds well to the fluorescence lifetime value of eGFP (between 2.4 and 2.6 ns[46]), suggesting that there is no influence on the fluorescence lifetime before oxidative stress when G3BP1 is diffusing almost freely in the cytoplasm. The clustering pattern observed for

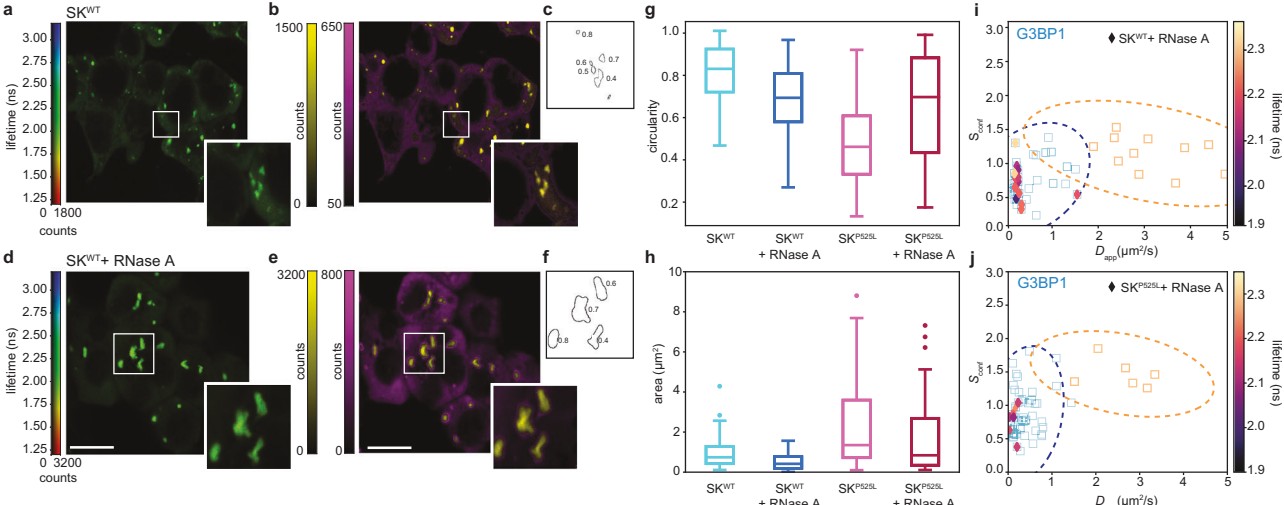

**Fig. 5 | Segmentation-based SG shape analysis. a** ISM reconstructed lifetime-based image of SK-N-BE WT cells expressing G3BP1-eGFP under oxidative stress before the RNAseA treatment. Bottom-right zoomed region in the white box. Scale bar 10 μm. **b** Phasor-based segmentation. ISM intensity images were obtained by back-projecting the points with a long lifetime (magenta) and a short lifetime (yellow). Scale bar 10 μm. **c** Outlines of the segmented SGs in the zoomed region. The numbers correspond to the circularity. **d** ISM reconstructed lifetime-based image of SK-N-BE WT cells expressing G3BP1-eGFP under oxidative stress during the RNAseA treatment. Bottom-right zoomed region in the white box. Scale bar 10 μm. **e** Phasor-based segmentation. ISM intensity images were obtained by back-projecting the points with a long (magenta) and a short lifetime (yellow). Scale bar 10 μm. **f** Outlines of the segmented SGs in the zoomed region. The numbers correspond to the circularity. **g** Box-plots of the circularity of the segmented SGs in SK-N-BE WT cells (left) and in mutated cells (right) before the addition of RNAseA and during the RNase treatment. The data are from two independent experiments (WT: $n = 106$ before RNAseA and $n = 72$ during RNAseA; mutated: $n = 51$ before RNAseA

and $n = 62$ during RNAseA). The horizontal line in each box represents the median, the edges are the 25th and 75th percentile, and the vertical line extends to the minimum and maximum data points. **h** Box-plots of the area of the segmented SGs in SK-N-BE WT cells (left) and in mutated cells (right) before the addition of RNAseA and during the RNase treatment. The data are from two independent experiments (WT: $n = 106$ before RNAseA and $n = 72$ during RNAseA; mutated: $n = 51$ before RNAseA and $n = 62$ during RNAseA). The horizontal line in each box represents the median, and the edges are the 25th and 75th percentile. The single points are the outliers. **i** Confinement strength in relation to the diffusion coefficient of G3BP1 (diamonds) during the RNAseA treatment in WT SK-N-BE cells overlaid to the clustered data (empty squared). The color depicts the fluorescence lifetime. **j** Confinement strength in relation to the diffusion coefficient of G3BP1 (diamonds) protein during the RNAseA treatment in mutated SK-N-BE cells overlaid to the clustered data (empty squared). The color depicts the fluorescence lifetime value. Source data are provided as a Source Data file.

confinement strength (see Fig. 3c, d) finds validation in the fluorescence lifetime information, which exhibits a similar clustering tendency. This correlation further establishes the connection between changes in fluorescence lifetime and alterations in protein dynamics (see Fig. 4h).

## Dual-color FCCS reveals proteins interaction

We conducted two-color cross-correlation spectroscopy (dcFCCS) measurements to investigate the interaction between G3BP1 and FUS during SG formation. Specifically, we correlate the signal of G3BP1-eGFP registered with the SPAD array detector with the signal of FUS-RFP registered with the single-element SPAD detector. Additionally, we performed ISM imaging for G3BP1 protein, and conventional confocal imaging for FUS protein. To ensure accurate detection volumes, we calibrated both the SPAD array detector and the single-element SPAD detector using fluorescent beads through circular scanning FCS (Supplementary Fig. 11). Circular scanning FCS allowed us to measure the detection volumes of diffusing particles without making assumptions about their diffusion coefficients (Supplementary Note 1). The detection volume of the single-element detector corresponds in dimension with the detection volume of the inner $3 \times 3$ elements of the SPAD array detector (about 330 nm).

In the context of WT cell conditions, cross-correlation measurements were less effective because the signal from one of the proteins, depending on the measured position, was often noise-dominated due to the distinct compartmentalization of the two proteins (Supplementary Fig. 12). Conversely, in mutated SK-N-BE cells, both FUS and G3BP1 are co-present in the cell cytoplasm before the stress condition

(Fig. 6a). Inducing oxidative stress is reflected by the formation of SGs in the cytoplasm, which are co-localizing with FUS$^{P525L+}$ condensates (Fig. 6b). In the pathologically mutated SK-N-BE condition, we can record the fluorescence signal from both proteins simultaneously and reconstruct the intensity time trace for both detectors, as well as the photon arrival-time histogram (Fig. 6c, d). Autocorrelation and cross-correlation of G3BP1 and FUS before and during stress are shown in Fig. 6e, f. The amplitude of the normalized cross-correlation function is null (Supplementary Fig. 12b, c) in the case of WT SK-N-BE cells (where FUS and G3BP1 are not co-localized). In SK-N-BE mutated cells, before the granule formation, G3BP1 and FUS both showed good autocorrelation curves. Surprisingly, the cross-correlation curve still can be calculated and fitted, indicating some degree of interaction between the two proteins on longer time scales. This might suggest the formation of few small oligomers between G3BP1 and mutated-FUS protein. Measurements inside SGs (Fig. 6f) show longer time decay on both autocorrelation curves and on the cross-correlation curve, indicating the formation of bigger condensates containing both proteins. The amplitude of the cross-correlation function is higher (gray curve in Fig. 6f) than before the stress, suggesting the formation of larger condensates and the increase in the interaction strength between the two proteins. This is also well represented by the relative amplitude of the cross-correlation curves (Fig. 6h) before (around 0.1) and after the formation of the SGs (around 0.4). The measured increase indicates a stronger interaction of G3BP1 and FUS inside SGs. The stronger interaction between FUS$^{P525L+}$ and G3BP1 inside the SGs of mutated cells confirms our findings of the SGs segmentation after the RNaseA treatment, wherein the mutated cell condition. There, the depletion of RNAs was less evident in the

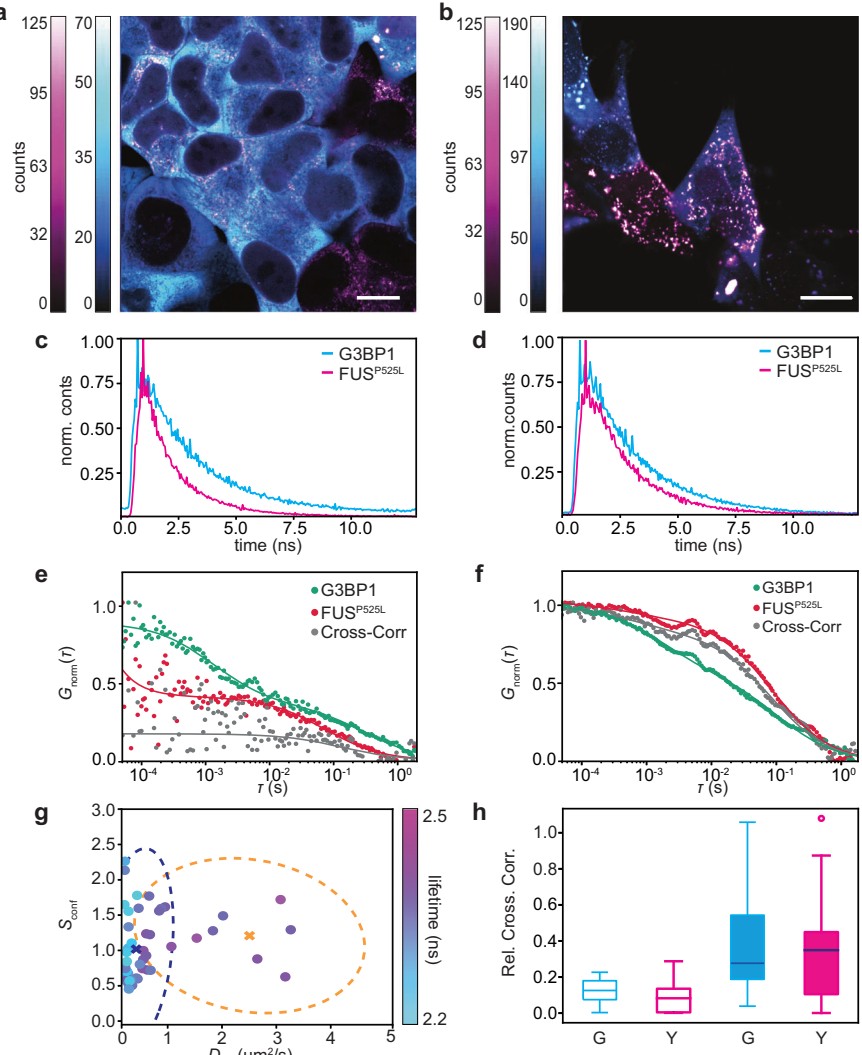

**Fig. 6 | Cross correlation fluorescence lifetime spectroscopy. a** Intensity-based ISM image of an SK-N-BE mutated cell expressing G3BP1-eGFP (cytoplasm, in cyan) and FUS^P525L+-RFP (cytoplasm, magenta) before the oxidative stress. Scale bar 7 μm. **b** Intensity-based ISM image of a SK-N-BE mutated cell expressing G3BP1-eGFP (cytoplasm, in cyan) and FUS^P525L+-RFP (cytoplasm, magenta) after the oxidative stress. Scale bar 5 μm. **c** Time decay histograms for G3BP1 (cyan) and FUS^P525L+ (magenta) in SK-N-BE mutates cells (**a**) before the oxidative stress. **d** Time decay histograms for G3BP1 (cyan) and FUS^P525L+ (magenta) in SK-N-BE mutates cells (**b**) during the oxidative stress. **e** Autocorrelations and cross-correlation curve (gray) calculated with the same data shown in (**c**) for G3BP1-eGFP (green), FUS^P525L+-RFP (red), in the cytoplasm before the SG formation. **f** Autocorrelations and cross-correlation curve (gray) calculated with the same data shown in (**d**) for G3BP1-eGFP (green), FUS-RFP (red) inside a SG. **g** Confinement strength and diffusion coefficient of G3BP1-eGFP in mutated SK-N-BE cells in relation to their fluorescence lifetime. The K-mean clustering algorithm has been applied to the datasets. The crosses represent the centroid position of each cluster. The dotted ellipses represent the covariance confidence level of each group (*n* = 32). **h** Relative cross-correlation amplitude. Empty box-plots represent measurements on SK-N-BE mutated cells before the stress (*n* = 10 from two independent experiments for both proteins). Filled box-plots represent measurements of SK-N-BE mutated cells after the stress. Cyan is the relative amplitude in relation to the G3BP1 signal, while the magenta is in relation to FUS (*n* = 20 from two independent experiments for both proteins). The horizontal line in each box represents the median, box edges are the 25th and 75th percentile, and the vertical line extends to the minimum and maximum data points. Source data are provided as a Source Data file.

structure of SGs in the mutated case compared to the WT case, indicating a strong protein-protein interaction able to counteract the decreased contribution of RNAs.

As we acquired the data to the single-photon level, we can always include information about the protein nano-environment, hence, the fluorescence lifetime. In a single dataset, we can finally combine cross-correlation with svFCS analysis and fluorescence lifetime, collecting a complete, highly informative set of physical parameters for studying biomolecular condensation. Similarly to experiments of the confinement strength on G3BP1, the data shows a similar clustering pattern (Fig. 6g, data clustered with k-mean algorithm), and two clusters were found, corresponding to protein in SGs and not condensate cytoplasmic protein.

## Discussion

We have introduced a comprehensive fluorescence-based approach to the in-depth investigation of biomolecular condensates in living cells, capable of spanning various spatial and temporal length scales. Our method integrates a conventional fluorescence laser-scanning microscope with an asynchronous read-out SPAD array detector, and a time-tagging DAQ module, enabling single-photon-resolved measurements. Remarkably, our single-photon fluorescence-based platform allows for high-resolution structural and functional imaging through time-lapse FLISM. Additionally, the platform seamlessly integrates various measurements, including fluorescence lifetime analysis, dcFCCS, and svFCS. For a given fluorescent biomolecule, this multi-scale and multi-parameter approach allows us to quantify the

molecular macro-environment (or sub-cellular environment), the nano-environment, the dynamics, the sub-diffraction environment (or mobility mode), and the interactions inside living cells.

We showed the potential of our method in investigating the formation of stress granules through liquid-liquid-phase separation. While conventional and super-resolved imaging offer valuable insights, they alone fall short of elucidating dynamics and interactions at the single-molecule level. To overcome these limitations, we showcased the capabilities of our integrated platform in quantifying the dynamics of specific RNA-binding proteins, such as G3BP1 and FUS, involved in the formation of SGs in SK-N-BE cells. We proved the viability of our method on both WT cells and mutated cells that carry one of the typical protein mutations of ALS patients.

While imaging and conventional FCS can reveal only partial information about the role of G3BP1 in the SG formation, our integrated approach provides novel insights to the best of our knowledge. Intensity-based imaging easily captures the increase and decrease of local concentration of G3BP1 during the three phases of the SG formation—pre-stress, post-stress, and recovery—as well as the structural shape of SGs. In parallel, conventional FCS shows changes in the diffusion coefficients of G3BP1 during the SG assembly and dissolution. However, it is through svFCS and confinement strength analysis that we validate the dynamic partitioning behavior of G3BP1, confirming the liquid-like nature of SGs. Notably, we observe similar G3BP1 dynamics in both WT and mutated cells. The advantages of our comprehensive fluorescence-based approach over conventional methods become even more evident when we study FUS dynamics. While conventional FCS unveils changes in diffusion coefficients during the three phases of SG formation, it cannot fully capture alterations in FUS protein dynamics in mutated cells upon stress induction. In contrast, svFCS reveals significant differences in confinement strength for mutated FUS protein before and after stress induction, indicating distinct underlying phase-separation processes. Comparing the results for G3BP1 and FUS, we note that the two proteins exhibit distinct dynamics when not stressed. However, once the two proteins are internalized into SGs, their diffusion coefficients and confinement strengths become strikingly similar, hinting toward an interaction process. We directly confirm this interaction by measuring the cross-correlation function between FUS and G3BP1 during the stress induction. The amplitudes of the cross-correlation functions increase in measurements inside the SGs, signifying a clear interaction between mutated FUS protein and G3BP1.

In addition to the insights provided by fluorescence fluctuation spectroscopy and intensity-based imaging, our approach combines these findings with the fluorescence lifetime analysis. Recently, the fluorescence lifetime has garnered the interest of various groups, being used as a reporter for functional changes inside living samples[36]. We observe changes in the fluorescence lifetime due to the protein compartmentalization after the formation of the SGs. We expect the fluorescence lifetime to be of great importance in studying protein interaction and structural changes, complementing its intrinsic metabolic information by using it to perform, for example, FLIM-FRET experiment. Once again, our integrated platform offers a simpler and more gentle approach to comprehensive measurement of biological samples during kinetic processes.

The interaction that we directly measured between mutated FUS and G3BP1 inside the SGs might be driven by RNA molecules, such as long non-coding RNAs (lncRNAs)[47–49], which are known to be present inside the SGs. In perspective, we envision our methods playing a crucial role in uncovering the primary involvement of RNA molecules in various neurodegenerative diseases characterized by the formation of insoluble aggregates, such as Alzheimer's and Parkinson's diseases. As an additional proof of the role of RNA molecules in SGs, we treated cells with RNaseA in order to deplete any RNA. Opposite to the formation of SGs, we found that here, svFCS and fluorescence lifetime

alone are not sufficient to study the overall process. In the WT cells, while the nano-environment, the dynamics, and the mobility modes are still similar to before the RNA depletion, the overall shape of SGs is affected by the presence of RNaseA. Thus, having a platform complemented with imaging possibilities is fundamental to comprehensively studying a biomolecular process. We found that RNA molecules have an important role in the compaction of the SGs but probably not in the molecular internal dynamics that protein-protein interactions might drive. The presence of FUS$^{P525L+}$ could imply a stronger protein-based interaction, which might influence the aberrant liquid-to-solid transition of SGs in the pathological condition. In general, our platform will be pivotal in directly monitoring the movement and the dynamics of single RNA molecules in the context of RNA and condensate-modifying therapeutics[50].

We anticipate a significant shift from conventional single-element detectors to SPAD array detectors in fluorescence laser-scanning microscopy. This shift will likely lead to the widespread adoption of the approach proposed in this study for understanding LLPS and uncovering other biomolecular mechanisms that drive essential cell functions. Due to the inherent variability of cellular processes, we firmly believe that only a comprehensive approach, combining multiple quantitative methods like the one presented here, will effectively help investigate the kinetics and mechanism behind them. We know that a dominant technology, such as the SPAD array detectors, which enables access to an extensive single-photon dataset, will allow for numerous additional analyses. We anticipate that our platform will enhance the information obtained during fluorescence spectroscopy and imaging measurements, fully compatible with any labeling, sample preparation, or biological process. Similar to how super-resolution microscopy has revolutionized structural imaging, we expect our method to transform the investigation of kinetic processes in living cells. Our comprehensive fluorescence-based platform empowers researchers to gain deeper insight into complex biological processes and molecular interactions at multiple scales.

With this vision in mind, we are actively contributing to the widespread adoption of the approach proposed in this study by sharing all the data analysis software presented in this work. This software will be integrated into a more general experimental environment, which includes our BrightEyes-TTM hardware[27], recently published by us as an open-hardware system for time-resolved measurements with SPAD array detectors.

## Methods

### Laser-scanning microscope

**Optical architecture.** To perform all the experiments, we modified a custom laser-scanning microscope designed for fluorescence fluctuation spectroscopy and imaging with SPAD array detectors[5,6,27]. Specifically, the microscope uses a $5 \times 5$ SPAD array fabricated with a 0.16 μm BCD technology[17] and cooled to −15 °C to reduce the dark-noise[6]. We additionally integrated a single-element SPAD detector ($PD-050-CTC-FC, Micro Photon Devices, Bolzano, Italy) to simultaneously register the fluorescence light within two different spectral windows i.e., to perform dual-color experiments. An overview of the SPAD array detector specifications is given in Supplementary Table 1. Briefly, the microscope excites the sample with two triggerable 485 nm and 561 nm pulsed laser diodes (LDH-D-C-485 and LDH-D-TA-560B, PicoQuant, Berlin, Germany). We combined the two beams with a dichroic mirror (F43-088, AHF Analysentechnik, Tübingen, Germany) and coupled them into a polarization-maintaining fiber before sending them to the microscope. The lasers are controlled by a pair of laser drivers, which can be synchronized via an oscillator module (PDL 828-L "SEPIA II", Picoquant). The oscillator also provides the synchronization signal to feed into the time-tagging module and to measure the photon-arrival times. The two co-aligned and collimated excitation beams are deflected by a pair of galvo-mirrors—to scan the probed

region across the specimen—and focused on the sample by using a 100×/1.4 numerical aperture objective lens (Leica Microsystems, Wetzlar, Germany). The emitted fluorescence signal is collected by the same objective lens, de-scanned by the two galvo-mirrors, and focused on the SPAD array detector or on the core of the single-mode fiber of the single-element SPAD detector. To spectrally separate the fluorescence between the two detectors, a short pass filter (F38-534, AHF Analysentechnik) is introduced in the optical path before the two detectors.

**Control and data-acquisition system.** We managed the microscope operations with an FPGA-based control unit (PRISM-Light control unit, Genoa Instruments, Genova, Italy) and the BrightEyes-MCS microscope control software. The microscope control unit drives the galvo-mirrors and the axial piezo-stage to position the probed region on different sample locations or scan the probed region across the whole sample. The control unit also records the signals from the SPAD (array and single-element detectors) in synchronization with the scanning beam system and provides—as outputs—the relative reference signals (i.e., pixel clock, line clock, and frame clock). The BrightEyes-MCS software is based on LabVIEW (National Instruments, Austin, TX) and Python and uses as a backbone the Carma application (Genoa Instruments)[51]. The BrightEyes-MCS provides a graphical user interface to control the microscope acquisition parameters (e.g., scanned region, pixel size, axial position, pixel dwell time), to register the digital signals of the SPADs, and to visualize the recorded signals (e.g., intensity images, time traces, and correlations) in real-time. In the context of fluorescence fluctuation spectroscopy, the BrightEyes-MCS software allows recording the fluorescence signal from each SPAD with a sampling (temporal bin) down to 500 ns. We used this acquisition modality to perform all experiments in which the fluorescence lifetime information is discarded: The photons registered by each SPAD are counted within the temporal bin to obtain the different intensity time traces. When we performed fluorescence lifetime analysis in combination with imaging or spectroscopy, we interleaved the BrightEyes-TTM card between the detectors and the control unit[27]: We connected all the SPAD outputs—transistor-transistor logic (TTL) signals—to the time-tagging module, together with the synchronization signal from the laser-driver and the reference signals from the scanning beam systems. The time-tagging module also replicates the input signals of the 3 × 3 central elements of the SPAD array detector and sends them to the control unit of the microscope for real-time visualization of the intensity signal. To differentiate the two data acquisition modalities, we named the first intensity-based measurement, i.e., we do not use the BrightEyes-TTM, and the second lifetime-based measurement, i.e., we register the single-photon arrival time with the BrightEyes-TTM.

### Sample preparation

**Calibration samples.** For calibrating the confocal volumes of our system, we used a solution of YG carboxylate fluoSpheres (REF F8787, 2% solids, 20 nm diameter, actual size 27 nm, exc./em. 505/515 nm, Invitrogen, ThermoFisher, Waltham, MA, USA) diluted 5000× in ultrapure water or a solution of Red carboxylate fluoSpheres (REF F8786, 2% solids, 20 nm diameter, exc./em. 580/605 nm, Invitrogen) diluted 3000× in ultrapure water.

**Alpha-synuclein purification and labeling.** Wild type $\alpha$-syn and $\alpha$-syn with the addition C141 ($\alpha$-syn$^{C141}$) were respectively purified as described in ref. [52]. Briefly, both proteins were recombinantly produced in E. coli. The pT7-7 aSyn C141 plasmid was a gift from Gabriele Kaminski Schierle (Addgene plasmid #108866; RRID: Addgene_108866). After lysis with boiling, ammonium sulfate precipitation, and dialysis, the proteins were subsequently purified via anion-exchange and size-exclusion chromatography with HiTrap™ Q and HiLoad™ 16/600 Superdex™ 75 columns (Cytiva, Marlborough, MA, USA). The cysteine

in position 141 was used to label $\alpha$-syn$^{C141}$ with Atto-488 via a maleimide reaction according to the manufacturer's instructions. Briefly, the dye is dissolved into water-free DMSO (Thermo Fisher) to a concentration of 10 mM. $\alpha$-syn$^{C141}$ dissolved in 20 mM phosphate buffer was incubated with 1.3 molar excess of Atto-488-maleimide (ATTO-TEC GmbH, Siegen, Germany) for 120 min at room temperature. After the incubation, the labeled protein is separated from the free Atto-488 molecules via PD-10 desalting columns (Cytiva). The tagged protein was immediately checked for concentration, aliquoted, frozen, and kept at −80 °C. $\alpha$-syn and free dye concentrations are measured by UV-vis absorption spectroscopy (Nanodrop ND-1000, Thermo Scientific Technologies).

**Plasmid construction and stable cell lines selection.** The transposable element vectors for inducible expression of RFP-FUS$^{wt}$ and RFP-FUS$^{P525L+}$ were described in ref. [40]. The plasmid for G3BP1-eGFP expression was cloned starting from an ePB-bsd-Eif1a (PiggyBac transposable vector) backbone[53] using sequential In-Fusion Cloning (Takara Bio Inc., Otsu, Japan) reactions. CloneAmp HiFi PCR Premix (Clontech, Mountain View, CA, USA) was used for all the PCR amplifications. To produce SK-N-BE(2) cell lines (gently provided by the group of Irene Bozzoni) stably expressing G3BP1-eGFP, $2.5 \times 10^5$ cells were plated on 6 cm dishes. The next day, cells were transfected with a solution of Optimem (Thermo Fisher), 5 μg of specific plasmid, 0.5 μg of hybrid transposase plasmid and Lipofectamine 2000 transfection reagent (Invitrogen) with a 1:2.5 DNA to transfection reagent ratio.

### FLISM and FLFS

**Experimental protocol.** For the experiments with fluoSpheres a droplet of fluoSpheres diluted in ultrapure water was poured on a coverslip; a fresh sample solution was prepared for each measurement. All measurements were performed at room temperature. For the experiments with $\alpha$-syn, the protein was previously filtered with 0.22 μM Millex Syringe filters (Durapore PVDF, Merck KGaA, Darmstadt, Germany) in order to exclude the presence of potential protein oligomers or aggregates. For the monomeric $\alpha$-syn in non-LLPS condition, 700 nM Atto488-maleimide $\alpha$-syn$^{C141}$ was diluted in 20 mM phosphate buffer and 100 mM KCl (Merck KGaA); for the LLPS experiment, Atto488-maleimide tagged $\alpha$-syn$^{C141}$ and not-tagged $\alpha$-syn$^{C141}$ were mixed with a ratio of 1:20 in the LLPS buffer (20% poly-ethylene glycol 8000 (PEG-8K, P1458, Merck KGaA) in 20 mM phosphate buffer). All measurements were performed at room temperature by dropping 100 μl in a $\mu$-Slide 8 well (Ibidi GmBH, Gräfelfing, Germany). For disruption of $\alpha$-syn LLPS, after 30 min LLPS was triggered, 1,3-Hexanediol (Merck KGaA), previously diluted in LLPS buffer, was added to the sample with a final concentration of 10%. For the experiments with SK-N-BE cells, cells were seeded onto a $\mu$-Slide 8 well plate (Ibidi GmbH) and imaged in Live-Cell Imaging Solution (Thermo Fisher Scientific) at 37 °C. Before each spectroscopy measurement, the cells were visually inspected by imaging. The axial position for the spectroscopy measurements was placed in the middle of the chosen cell. Multiple planar positions in cells were selected to probe different points (in the cytoplasm or inside SGs) at different temporal points of the process. To induce the oxidative stress the cells were treated with 0.5 mM of sodium arsenite (Merck KGaA). In the experiments with the addition of RNaseA, cells were permeabilized with Triton X-100 0.05% (Merck KGaA) for about 5 min prior to the addition of RNaseA 0.2 mg/ml (Merck KGaA)[44]. The cells were then investigated for about 30 min. The fluorescence intensity was acquired for each measured position for about 120 s and analyzed offline. All measurements were performed at 37 °C inside a temperature-controlled chamber (Bold Line Temperature Controller, Okolab, Sewickley, PA, USA).

**Image reconstruction and analysis.** We reconstructed the ISM images with the adaptive pixel-reassignment method[25]. This process

involves several steps: From the list of photons provided by the BrightEyes-TTM, we reconstructed the 4D photon counting image/histogram ($ch, x, y, \Delta t$), where $ch$ represents the detector element, and $x$ and $y$ denote the sample position (or pixel). $\Delta t$ is the single-photon arrival time. We then integrated the 4D image along the $\Delta t$ dimension. Next, we applied a phase-correlation registration algorithm to align all the images ($x, y|ch$) with respect to the central image. We stored the resulting lateral shifts, for each channel, in a shift-vector fingerprint. Following that, we integrated the registered image along the $ch$ dimension to obtain the ISM intensity-based images. To obtain the lifetime-based ISM image, we followed these steps: Starting from the 4D data set ($ch, x, y, \Delta t$), we used the shift-vector fingerprint for each $\Delta t$ value to shift the relative 2D image. We then integrated the result along the $ch$ dimension. Using the resulting 3D data set ($x, y, \Delta t$) and the ImageJ plugin FLIMJ[54], we obtained the ($x, y, \tau_{fl}$) image. In particular, we fitted the arrival time histogram of each position/pixel with a single-exponential decay model. Alternatively, we applied phasor analysis to the same 3D ($x, y, \Delta t$) dataset. Phasor analysis allowed us to interpret the fluorescence lifetime by projecting the photon arrival-time histograms in a 2D coordinate system without the need for fitting. For each position/pixel, we calculated the phasor coordinates ($g, s$) using cosine and sine summations[55,56]. To avoid artifacts, we performed the MOD mathematical operation of the time-correlated single photon counting histograms with the laser repetition period value. To account for specific but fixed delays introduced by each SPAD element, we calibrated the system by measuring the instrument response function of the complete setup (microscope, detector, and BrightEyes-TTM) using a solution of Atto-495 (with $\tau_{fl} = 1\,ns$) or Atto-594 (with $\tau_{fl} = 3.9\,ns$). Additionally, we used Fiji for SG shape and area analysis[57]. Circularity was defined as $4\pi(\text{area/perimeter}^2)$, where a circularity equal to 1 corresponds to a perfect circle.

**Fluorescence correlation calculation and analysis.** We calculated the correlations directly on the lists of absolute photon times[27,58]. For the sum $3 \times 3$ and sum $5 \times 5$ analysis, the lists of all relevant SPAD channels were merged, and the correlations were calculated. The data was then split into chunks of 10 or 5 s, depending on the probe observed, and for each chunk, the correlations were calculated. The individual correlation curves were visually inspected, and all curves without artifacts, introduced by cell movements or bleaching, were averaged. For both single-point and circular scanning[59] FCS, the correlation curves were fitted with a 1-component model assuming a Gaussian detection volume, as described in ref. 6. The theoretical fitting functions for FCS are briefly summarized in Supplementary Note 1. For the circular FCS measurements[59], the periodicity and radius of the scan movement were kept fixed while the amplitude, diffusion time, and focal spot size were fitted. This procedure was used for the fluorescent beads and allowed calibrating the different focal spot sizes (i.e., central, sum $3 \times 3$ and sum $5 \times 5$, respectively, 276, 330, and 390 nm). For the conventional FCS measurements, the focal spot size was fixed at the values found with circular scanning FCS, and the amplitude and diffusion times were fitted. Since we approximated the PSF as a 3D Gaussian function with a $1/e^2$ lateral radius of $\omega_0$ and a $1/e^2$ height of $k \times \omega_0$ (with $k$ the eccentricity of the detection volume, $k = 4.5$ for the central element probing volume, $k = 4.1$ for the sum $3 \times 3$ and sum $5 \times 5$ probing volume), the diffusion coefficient $D$ can be calculated from the diffusion time $\tau_D$ and the focal spot size $\omega_0$ via $D = \omega_0^2/(4\tau_D)$.

**Statistics and reproducibility**
All experiments were performed independently at least two times. Fluorescence ISM, FLISM, phasor plots, svFCS and fluorescence decay histogram representative images are shown. Welch's $t$-test with a confidence interval of 0.95 was used to analyze the differences between two groups. The results are presented as means and standard deviations, and $p$ values < 0.05 were considered statistically significant. In svFCS experiments, the time traces were split into chunks of 10 or 5 s, depending on the probe observed, and for each chunk, the correlations were calculated. The individual correlation curves were visually inspected, and all curves without artifacts, introduced by cell movements or bleaching, were averaged. There was no estimate of variation within each group of data. The experiments were not randomized. The Investigators were not blinded to allocation during experiments and outcome assessment.

**Reporting summary**
Further information on research design is available in the Nature Portfolio Reporting Summary linked to this article.

## Data availability
The single-photon data shown in Fig. 2 of this manuscript have been deposited in Zenodo repository under the accession code https://doi.org/10.5281/zenodo.10046694. The datasets for the cell experiments are large and, as such, they can be made available by the corresponding author upon request; full access can be obtained by request to the corresponding author. Requests will be fulfilled within 2 of weeks. Source data are provided as a Source Data file.

## Code availability
The source code for analyzing the photon time-tagging measurements (fluorescence lifetime fluctuation spectroscopy and fluorescence lifetime image scanning microscopy) has been deposited in the GitHub BrightEyes-TTM repository as a part of a larger open-hardware/software project which aims at democratizing single-photon laser-scanning microscopy based on SPAD array detector[27] and it is available also on Zenodo[60] https://doi.org/10.5281/zenodo.7064910. The source code for analyzing the intensity-based measurements (fluorescence fluctuation spectroscopy) has been deposited on the GitHub BrightEyes repository (https://github.com/VicidominiLab/BrightEyes-TTM/tree/v2.0/scriptsFLFS).

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

## Acknowledgements

This project has received funding from: the Fondazione San Paolo, "Observation of bio-molecular processes in live-cell with nano-camera", No. EPFD0098 (E.S., S.Z., and G.V.) and, "Augmented fluorescence correlation spectroscopy with a novel SPAD array detector to observe complex biological processes in living cells", Trapezio No. 71100 (E.P.); the European Research Council, "BrightEyes", ERC-CoG No. 818699 (G.V.), "ASTRA", ERC-SyG No. 855923 (G.G.T. and I.B.); the European Innovation Council, "ivBM-4PAP", Pathfinder No. 101098989 (G.G.T.); the European Union's Horizon 2020 research and innovation program under the Marie Sklodowska-Curie grant agreements, "SMSPAD", No. 890923 (E.S.) and "MINDED" No. 754490 (E.Z.); the NextGeneration EU PNRR MUR—M4C2—Action 1.4—Call Potenziamento strutture di ricerca e creazione di "campioni nazionali di R&S" (CUP J33C22001130001), "National Center for Gene Therapy and Drugs based on RNA Technology", No. CN00000041 (G.G.T., I.B, and G.V.); by Associazione Italiana per la Ricerca Sul Cancro, "Circular RNAs: novel players and biomarkers in tumorigenesis", IG 2019 No. 23053 (I.B.); by Ministero dell'Istruzione, dell'Università e Della ricerca (MIUR), "Non-coding RNAs, new players in gene expression regulation: studying their role in neuronal differentiation and in neurodegeneration", PRIN2017 No. 2017P352Z4 (I.B.). We thank: all members of the Molecular Microscopy and Spectroscopy group and the RNA Systems Biology group at the Istituto Italiano di Tecnologia (IIT) for the daily discussion and suggestions about the project; Prof. Alberto Diaspro and Dr. Paolo Bianchini (Nanoscopy & NIC@IIT, IIT) for valuable discussions; Dr. Michele Oneto (Nikon Imaging Center, IIT) for support on the experiments; Prof. Alberto Tosi, Prof. Federica Villa, Dr. Mauro Buttafava (Politecnico di Milano), Dr. Marco Castello, and Dr. Simon-luca Piazza (Genoa Instruments) for the realization of the single-photon-avalanche-diode detector array; all members of the RNA Initiative at the Istituto Italiano di Tecnologia, Prof. Stefano Gustincich (Non-coding RNAs and RNA-based therapeutics, IIT) and Dr. Francesco Nicassio (Genomics Science, IIT) for their contribution to the long-term vision of this project.

## Author contributions

G.V. conceived the idea. E.P., E.S., and G.V. developed the methodologies for the study. E.Z., G.G.T., I.B., and G.V. supervised and coordinated the project. F.C., D.M., and E.V. prepared the living cell sample. S.Z., J.R., and E.Z. prepared the in vitro sample. E.P., S.Z., and F.C. performed the living cell experiments. E.P., S.Z., and J.R. performed the in vitro experiments. E.P., S.Z., and G.V. analyzed the living cell and in vitro experiments. E.S. and G.V. designed and built, with the help of E.P., the custom laser-scanning microscope. E.S. designed and implemented the microscope control software. E.P. and E.S. designed and implemented the data analysis software. All authors discussed the results. E.P. and G.V. wrote the manuscript with input from all authors.

## Competing interests

G.V. has a personal financial interest (co-founder) in Genoa Instruments, Italy. G.V. has any non-financial interests. The remaining authors declare no competing interests (financial or non-financial).
