## [Peer Review File · Nature Communications]

REVIEWERS' COMMENTS

Reviewer #1 (Remarks to the Author)

The manuscript entitled « Content-enriched fluorescence lifetime fluctuation spectroscopy to study bio-molecular condensate formation » and written by Perego et al. proposes a methodological framework to investigate the spatiotemporal processes of biomolecule. This approach was applied to study the condensates appearing after oxidative stress. The manuscript is clearly written and the biological results are convincing and satisfactorily detailed. However, as mentioned by the authors, the methodology and the technical details of the methods used in this publication have already been published previously: ref. 5 (Slenders et al., *Light Sci Appl*, 2021) for spot variation FCS and FCCS, ref. 21 (Castello et al, *Nature Methods*, 2019) for ISM and FLIM, ref. 17 (Rosetta et al., *Nature Communications*, 2022) for fluorescence lifetime fluctuation spectroscopy. In other words, the work presented in this publication is a biological application of the different technics that have been published previously by the same group. I think that it is crucial to write clearly this point in the abstract in order to not confuse the readers of *Nature Communications*.

I have also other issues that have to be addressed:

- 1) Page 4, right column: the authors write: "The fluorescence lifetime of the tagged molecule is extracted by fitting the decay function." But they do not describe the results that they obtained and do not discuss the interest of this measurement.
- 2) Page 5, left column: the authors write: "the correlation curves show a second slower diffusing component in the fits (top-right)." I am not sure to understand the authors. If they speak about the correlation curve in figure 3 b), I am afraid that this interpretation is difficult to follow for the readers. I guess that the authors mean that the FCS curves have been fitted with a two-component model.
- 3) Page 5, right column: could the authors indicate how they calculate the apparent diffusion coefficient (especially in the case of a two-component model) ? Unless I am mistaken, I did not find information about it in the manuscript and this parameter was used also in figures 3, 4 and 5.
- 4) Page 6, figure 3 a) and b): Is it possible to show several FCS curves for the same biological condition in order to have an idea about the variability and standard deviation in living cells. I guess that the FCS curves shown in figure 3 a) and b) correspond to a single measurement of 120 s. Is it true ? If yes, it is possible that the cell has moved during the acquisition. Do the authors have compensated for that ?
- 5) Page 6, figure 4 c): the authors show the phasor plots of the stressed cell; Is it possible to show also the phasor plot for non-stressed cell for comparison ? The shift in lifetime should also be visible in this phasor plot.
- 6) Page 5, left column: the authors write: "we quantify the fluorescence lifetime by both fitting the photon arrival-time histograms for each pixel with a single-component exponential decay function(Fig.4a-b top- right) and by phasor analysis (Fig.4c-d)." However, a single exponential decay function should be characterized in the phasor plot by a spot localized on the semi-circle (and not inside the semi-circle as here). From the phasor plot analysis, it appears clearly that the decay is not a single component one. I think that it is necessary to analyze the lifetime data with a multi-component exponential function.
- 7) Page 6, figure 4 h): I am quite confused with this figure. When the lifetime is small (dark color) and the apparent diffusion coefficient is small, it corresponds to the stressed G3BP1 mutated cells. And when the G3BP1 mutated cells are stressed, the confinement strength decreases in comparison with the non-stressed cells (see figure S4a). However, this is the opposite in this figure 4. The authors should clarify this point.
- 8) Page 8, figure 5: the authors demonstrate convincingly that there is some cross-correlation between G3BP1 labeled with eGFP and FUS labeled with RFP after the oxidative stress, meaning that G3BP1 and FUS are diffusing together. Furthermore, the lifetime of G3BP1 is clearly decreasing after stress and eGFP and RFP are good donor and acceptor for FRET experiments. So, my question is: why the authors do not consider that all these results could be explained by the fact that there is some

FRET between G3BP1 and FUS ?

9) Page 11: For the analysis of FLISM data, the authors use an ImageJ plugin and apply a single exponential decay model. Usually, for improving the fitting, it is necessary to use the instrument response function (IRF). Do the authors have acquired an experimental IRF ? It could be interesting to see this IRF because it could be dependent on the SPAD array pixel and on the emission wavelength.

10) Supp. Material, figure S1: b) the FCS curve in red (monomeric alpha synuclein) is not well fitted with a single component model. Why the authors do not consider a two components model ? c) Is it possible to add the diffusion coefficient for monomeric alpha synuclein and also for the first component of LLPS and 10% 1,6-hexanediol.

11) Supp. Material, figure S6: Why the diffusion coefficients of the YG beads and the red beads which have the same diameter (20 nm) are so different (more than two orders of magnitude) ? Normally, the diffusion coefficient should be only related to the size of the bead.

Minor issues

1) Page 2, left column: "it is able to reveal also fast-scale (in the range of the measurement time, i.e., tens of seconds)". I understand the explanation of the authors but the term fast scale may be not well adapted to describe a process of tens of seconds.

2) In figure 2 f) and h), please add some details about the pixels used for obtaining the cross-correlation curves. Is it a single element or a sum ? Is it possible also to enlarge these 2 graphs because it is hard to visualize clearly the grey curves.

3) Page 5, left column: please correct the sentence: "Consequently, the cross-correlation also curves have low amplitudes."

4) Page 5, left column: please rephrase this sentence: "If, on the other hand, we measured only yellow-green fluorescent beads but with a 50-50 beam-splitter in the detection path and with green detection filters on the single-element detector, we can retrieve both good autocorrelation curves and a cross-correlation curve with high amplitude."

5) Page 7, right column: please correct the sentence: "By a simple centroid-based segmentation based on..."

6) Page 8, figure 5: "d Time decay histograms for G3BP1 (cyan) and FUS (magenta) in SK-N-BE mutates cells (d)". I think that it should be (b) instead of (d).

7) page 11, left column: "SK-N-BE cell lines stabling expressing...". Please replace "stabling" by "stably".

8) Page 11, left column: please check the parenthesis of the sentence: "For the monomeric α -syn in non-LLPS condition, 700 nM Atto488-maleimide α -synC141) was diluted in 20 mM phosphate buffer and 100 mM KCl (Sigma-Aldrich); for the LLPS experiment, Atto488-maleimide α -synC141) and wild-type α -synC141) were mixed with a ratio of 1:20 in the LLPS buffer (20% poly-ethylene glycol 8000 (PEG-8K, P1458, Sigma-Aldrich) in 20 mM phosphate buffer."

9) Page 11, right column: Is it possible to indicate the value of the beam waist used ? Usually, the calibration of the FCS measurements is performed on a solution of known diffusion coefficient, in order to fit only the amplitude and the focal volume. In this work, the authors fit three parameters: amplitude, diffusion time and focal volume. Why do they use this procedure that is known to lead to more variability ?

10) Supp. Material, page 1, equation S10: The introduction of the factor 0.35 is not explained. Where does it come from ?

11) Supp. Material, figure S2: the authors write: "b Dual-color Fluorescence Cross-Correlation Spectroscopy measurement of red fluorescent nanospheres in water." I am quite confused. Is it red or orange fluorescent beads ? and after: "equipped with the orange filter". Is it red or orange filter ?

12) Supp. Material, figure S5: Is it possible to show the distribution of the confinement strength and the apparent diffusion coefficient, in order to compare more easily with the results of figure S3 and S4 ?

13) Supp. Material, figure S6: "The detecting volume for this measurement is". Please finish this sentence.

Reviewer #2 (Remarks to the Author)

In this manuscript, the authors propose a multi-modal microscopy framework that, first, uses single photon avalanche diode (SPAD) arrays as an alternative method for implementing spot-variation Fluorescence Correlation Spectroscopy (svFCS). Second, they also add a single SPAD detector in conjunction with the SPAD array to the microscope setup in order to perform dual-color Fluorescence Cross-Correlation Spectroscopy (dcFCCS) experiments. This framework allows usage of a single dataset obtained using the whole platform for performing many different types of quantitative analyses to extract dynamical information across spatial and temporal scales. The authors demonstrate in detail the applicability of their framework by observing stress-granule (SG) formation and protein dynamics under their microscope and then analyzing the obtained datasets.

Existing svFCS techniques modify the size of the microscope's back-aperture to change the observation (or light integration) volume while collecting photons using a single detector. On the contrary, the proposed framework uses a detector array (5×5) to collect the photons. The integration is then done as a post-processing step where the observation volume or equivalently the number of detectors in the array to be used for integration is a user choice.

The proposed framework seems to have clear advantage over the existing svFCS methods in terms of temporal resolution, as dynamical information is being collected over different observation volumes simultaneously unlike other methods where information over different volumes is collected sequentially.

The manuscript is well-written and the scientific contribution is significant given that the individual techniques presented here were developed by the same authors and published earlier in Nature Communications (13, Article number: 7406 (2022)) and Nature Light (10, Article number: 31 (2021)). I recommend publication of this work in Nature Communications with the following comments addressed.

Comments:

1. A general comment for the authors would be that, given the wider audience of Nature Communications which includes both theorists and experimentalists, it would improve the readability of the paper if they define technical phrases clearly or use more descriptive phrases. For example: In the second paragraph on page 1, I am not sure what "sub-resolution" is referring to. Is it sub-diffraction limit or sub-camera pixel size? Similarly, I am not sure what the phrase "Content-Enriched" in the manuscript title is referring to. I have typically seen such phrases being used for describing deep-learning based computational microscopy algorithms that take into account prior information about a sample. For instance: see this paper "<https://www.nature.com/articles/s41592-018-0216-7>" published in Nature Methods about a technique called Content-aware image restoration (CARE). Therefore, it would be useful to either clarify such phrasing in the manuscript or come up with a different title to avoid any confusion.
2. In panel c of Fig. 2, I suspect the x-axis variable should be ω_0^2 as ω_0 should have units of length scale according to the FCS formulation provided in the Supplementary.
3. It would also be useful to explain in the manuscript why a linear relationship is expected in the τ_D vs ω_0^2 plots in Figures 2 & 3, especially when biomolecules being observed are not freely diffusing. From the FCS theory presented in the Supplementary, I can understand that diffusion time is directly related to the observation beam parameter ω_0^2 but it improves readability if it is explained in the main manuscript as well. Naively, one may think that, for heterogeneous samples, behavior may be different at large length scales as compared to shorter length scales such

that a linear relationship is not followed. In fact, the authors define the confinement strength S_{conf} to parametrize this exact phenomenon.

4. The definition of confinement strength S_{conf} is not clear to me. Which of the two terms in the ratio form the numerator and the denominator?

5. For the dsFCCS modality, I do not understand why the authors are using a combination of a single-point SPAD and a 5×5 SPAD array instead of two 5×5 SPAD arrays to obtain equal amount of spatio-temporal information for the two colors.

Reviewer #3 (Remarks to the Author)

(1) Dead time is an important factor for SPAD or SPAD array, which can distort the photon counting histogram, especially when imaging with high flux dynamic range. What are the parameters of these two used SPAD detectors? and how do the authors deal with it in the experiments.

(2) Fill factor is also an important factor for SPAD array, which affects the collection efficiency of echo photons or the lateral resolution of imaging obtained by SPAD array directly. Please give some information about this, such as the pixel pitch, the size of the active area of SPAD pixel in the array

(3) The dark count rate of the SPAD array detector should also be described, which is an important noise source for the imaging system. In particular, usually the dark counts of each pixel in the SPAD array cannot be completely consistent, and even hot pixels with a very high dark count rate may appear.

(4) P.9 : 'We additionally integrated a single-element SPAD detector (SPD-050-CTC-FC, Micro Photon Devices, Italy) to register simultaneously the fluorescence light within two different spectral windows to perform dual-color experiments.'

Why use two different SPAD to perform dual-color experiments? One SPAD array, and one single-element.

What is the advantage over a two-color system built on two unit SPAD detector? or a two-color system built on two SPAD array?

(5) P.10: 'The time tagging module also replicates the input signals of the 3×3 central elements of the SPAD array detector and sends them to the control unit of the microscope.'

Is it the SPAD array is used as a single element detector with its 3×3 central elements as one detector element?

Is there any difference between the usage of SPAD array in this manuscript and that in Ref [1,2]? which use SPAD array to realize super-resolution microscopy imaging.

[1] Tenne R, Rossman U, Rephael B, et al. Super-resolution enhancement by quantum image scanning microscopy[J]. Nature Photonics, 2019, 13(2): 116-122.

[2] Lubin G, Tenne R, Antolovic I M, et al. Quantum correlation measurement with single photon avalanche diode arrays[J]. Optics Express, 2019, 27(23): 32863-32882.

(6) Page 10: 'the BrightEyes-MCS software allows recording the fluorescence signal from each SPAD with a sampling (temporal bin) down to 500 ns. '

Is 500 ns the lowest integral time for intensity based measurement?

Reviewer #4 (Remarks to the Author)

Paper by Perego et al applies their recently developed FLIM,FCS-based technique (published previously; BrightEyes-TTM) using the power of mulθ-array SPAD (single-photon avalanche diode) detector to study LLPS formaθon and dynamics of stress granules(SG)-linked proteins G3BP1 and FUS in cells. Specifically, they were able to determine differences in diffusion coefficients of SG-linked proteins in stressed (with 0.5mM arsenite) and not stressed cells, as well as mutant FUS P525L, and differences in spaθal dynamics behavior by invoking the Sconf (confinement) parameter. Overall, the use of new combined Technologies is good, it is not clear what new insights was gained with the new technique that was not known previously.

Major Comments

1. It is difficult to assess the advantage of the mulθ-SPAD detector vs convenθonal single SPAD or other fluorescence methods because the conclusions such as slower movements of stressed proteins can be obtained with convenθonal FLIM, FCS or FRAP technique. Figures or data that shows criθcal technical advantages are needed, e.g. temporal resoluθon advantages; ie., what fast θmescales was captured that couldn't be captured previously; spaθal resoluθon inaccessible previously,etc.
2. Where are the FCS spots in the cells? Is there correlaθon between the intensiθes of the spots, or the concentraθon vs the diffusion coefficients or Sconf? Also, it is not clear how significant is the cluster differences (Fig. 3c,d) between the stress and the not stressed, and mutants? Or is it just heterogeneity of the system.
3. The nano-environment could be a big advantage for the technique that could have been more explored beζer to provide more substanθal insights on the formaθon of the stress granules, for examples, there is literature report that SGs have a dynamic outer shell and a more solid core. Can they provide this temporal and spaθal resoluθon? If they add rnase to remove RNA in SGs, can they see solidificaθon, changes in the spaθo-temporal dynamics and nano-environment. They have also explored only heterotypic interacθon between G3BP and FUS, how about the homotypic FUS, FUS cross correlaθon, which is relevant for irreversible aggregaθons?

Minor Comments

There are some spelling or grammatical errors, such as 'manufacture' fig. 2 legend, 'sodium- arsenite' shouldn't have a dash.

Reviewer #1 (Remarks to the Author):

The manuscript entitled « Content-enriched fluorescence lifetime fluctuation spectroscopy to study bio-molecular condensate formation » and written by Perego et al. proposes a methodological framework to investigate the spatiotemporal processes of biomolecule. This approach was applied to study the condensates appearing after oxidative stress. The manuscript is clearly written and the biological results are convincing and satisfactorily detailed. However, as mentioned by the authors, the methodology and the technical details of the methods used in this publication have already been published previously: ref. 5 (Slenders et al., Light Sci Appl, 2021) for spot variation FCS and FCCS, ref 21 (Castello et al, Nature Methods, 2019) for ISM and FLIM, ref. 17 (Rosetta et al., Nature Communications, 2022) for fluorescence lifetime fluctuation spectroscopy. In other words, the work presented in this publication is a biological application of the different technics that have been published previously by the same group. I think that it is crucial to write clearly this point in the abstract in order to not confuse the readers of Nature Communications.

We sincerely appreciate the thoughtful review provided by the referee and their recognition of the biological significance of our manuscript. We are grateful for their constructive feedback, which has greatly contributed to enhancing the quality of our work.

We have diligently addressed all the points raised by the referee. Notably, we have revised the abstract and incorporated additional paragraphs in the introduction to offer a clearer perspective on the primary objective of our manuscript. Our central aim is to show that single-photon microscopy can provide a comprehensive tool for studying biomolecular processes.

We acknowledge that many of the SPAD array-based methods we employ have been previously demonstrated by our group. However, it is essential to emphasize that previous works primarily served as proof-of-concept studies. In our current work, we have taken a significant step forward by demonstrating the comprehensive use of these methods, alongside the introduction of new techniques such as fluorescence cross-correlation spectroscopy (FCCS), all within a single measurement and for a practical application such as understanding the biomolecular condensations process.

We believe that this advancement is particularly noteworthy as it offers several key benefits, including accelerated measurements, reduced photo-bleaching, and a holistic view of biological processes. This stands in contrast to the sequential execution of techniques like spatiotemporal fluctuation analysis (svFCS), fluorescence lifetime imaging microscopy (FLIM), and FCCS.

To underscore the effectiveness of our proposed framework for studying biomolecular condensation processes, we are pleased to report that this framework has already been applied in experiments for two research papers currently under review.

We have marked the changes made in red within the manuscript to facilitate the referee's review process. Additionally, we provide a detailed point-by-point response to the referee's comments below.

Comments:

1) Page 4, right column: the authors write: “The fluorescence lifetime of the tagged molecule is extracted by fitting the decay function.” But they do not describe the results that they obtained and do not discuss the interest of this measurement.

We added this section in the manuscript: “As fluorescence lifetime is influenced by the surrounding environment, this parameter is often used as a reporter of the nano-environment of the fluorophore. As a proof-of-principle, we quantified the fluorescence lifetime value for the YG fluorescent beads, obtaining a value of (1.3 ± 0.1) ns.”

2) Page 5, left column: the authors write: “the correlation curves show a second slower diffusing component in the fits (top-right).” I am not sure to understand the authors. If they speak about the correlation curve in figure 3 b), I am afraid that this interpretation is difficult to follow for the readers. I guess that the authors mean that the FCS curves have been fitted with a two-component model.

As stated by the reviewer, the FCS curves have been fitted with a two-component model. We added an arrow to better illustrate the slower fitted component in the autocorrelation curve of Fig. 3b top in comparison with Fig. 3a top. Some of the measurements in the SGs show a strong two-component trend, probably due to small oligomers and interaction forming during the process. To highlight the variability of the process we added an extra supplementary figure (Fig. S3), where more autocorrelation curves (only correlation from the Sum 5x5 signal is shown) from different cell positions are shown.

3) Page 5, right column: could the authors indicate how they calculate the apparent diffusion coefficient (especially in the case of a two-component model)? Unless I am mistaken, I did not find information about it in the manuscript and this parameter was used also in figures 3, 4 and 5.

We considered the apparent diffusion coefficient as the one measured considering all surface of the SPAD array detector (Sum 5x5). The signal coming from all the SPAD array detector channels is integrated, and the autocorrelation is calculated on the summed signal.

As reported here, “The apparent diffusion coefficients, i.e. the ones measured with the whole detector area”.

4) Page 6, figure 3 a) and b): Is it possible to show several FCS curves for the same biological condition in order to have an idea about the variability and standard deviation in living cells. I guess that the FCS curves shown in figure 3 a) and b) correspond to a single measurement of 120 s. Is it true? If yes, it is possible that the cell has moved during the acquisition. Do the authors have compensated for that?

Yes, the single curves showed in Fig. 3 a-b are related to a single point measurement. The data are acquired for 120 s, the intensity trace is split in chunks of 10 s or 5 s and for each chunk the autocorrelation curve is calculated. The ones showing artefacts, related to cell movement or big aggregates, are deleted and the remaining ACFs are averaged. The curves in Fig. 3 a-b are the averaged correlations in one single point.

Here we show 11 chunks of 10 s each from the measurement of Fig.3a. The first 10 s have been discarded for an artifact passing (chunk 0, not shown here for clarity). As over time the autocorrelation curves are constant we averaged them. This is a sign that the cell didn't move during the measuring time.

We did add a supplementary figure (S3) showing the FCS curves for the same biological condition in different cells (top cells before oxidative stress, bottom cells under oxidative stress). We found that cells under stress shows a higher value of the intercept in the svFCS measurements (bottom right) compared to before the stress. Possibly, before the stress, the mobility of G3BP1 is depending by the position within the cell cytoplasm, and its variability is a reflection of the inhomogeneity of the cytoplasmic environment. On the other hand, the SG environment is more uniform and constrain the proteins to a lower mobility.

5) Page 6, figure 4 c): the authors show the phasor plots of the stressed cell; Is it possible to show also the phasor plot for non-stressed cell for comparison? The shift in lifetime should also be visible in this phasor plot.

We created a new supplementary figure with the phasor plot of non-stressed cells (Fig. S9). Both the fluorescence-based and the intensity-based ISM image show uniform signals in the cytoplasm. A single component of fluorescence lifetime is an indicator of the presence of a uniform biomolecule population (in this case monomeric cytoplasmic G3BP1-eGFP for non-stressed cells). This is confirmed by the phasor plot, which is centered on the universal circle. When the cells are under stress, the phasor plot is no longer centered on the universal circle, reflecting the shift in lifetime measured also by the combination of fluorescence lifetime and svFCS. The phasor cloud is larger compared to the one in Fig.4 as it has a lower signal-to-noise ratio.

6) Page 5, left column: the authors write: “we quantify the fluorescence lifetime by both fitting the photon arrival-time histograms for each pixel with a single-component exponential decay function (Fig.4a-b top- right) and by phasor analysis (Fig.4c-d).” However, a single exponential decay function should be characterized in the phasor plot by a spot localized on the semi-circle (and not inside the semi-circle as here). From the phasor plot analysis, it appears clearly that the decay is not a single component one. I think that it is necessary to analyze the lifetime data with a multi-component exponential function.

We thank the reviewer for the spotted inaccuracy. We did try a two-component fit with FLIMJ however, as the two possible components (SGs and cytoplasm) have a similar value of lifetime (2.2 ns and 2.5 ns respectively), the fitting algorithm performed worse compared to only 1 component ($\chi^2 = 0.15$ for 1-comp and 0.7 for 2-comp, data from ImageJ). For this reason, the FLISM images show an average lifetime value for each single pixel.

7) Page 6, figure 4 h): I am quite confused with this figure. When the lifetime is small (dark color) and the apparent diffusion coefficient is small, it corresponds to the stressed G3BP1 mutated cells. And when the G3BP1 mutated cells are stressed, the confinement strength decreases in comparison with the non-stressed cells (see figure S4a). However, this is the opposite in this figure 4. The authors should clarify this point.

We apologize for any confusion regarding Figure 4h. This figure illustrates the fluorescence lifetime of a stress-recovery experiment. During the recovery experiment, we observed a substantial variation in the confinement strength. To address this variability and provide a more comprehensive view of the confinement strength and diffusion coefficient values during the recovery experiment, we have introduced an additional supplementary figure, Fig. S7. This variation is also observable in Fig. S6. We hope that the inclusion of these figures will help clarify the observed dynamics.

8) Page 8, figure 5: the authors demonstrate convincingly that there is some cross-correlation between G3BP1 labeled with eGFP and FUS labeled with RFP after the oxidative stress, meaning that G3BP1 and FUS are diffusing together. Furthermore, the lifetime of G3BP1 is clearly decreasing after stress and eGFP and RFP are good donor and acceptor for FRET experiments. So, my question is: why the authors do not consider that all these results could be explained by the fact that there is some FRET between G3BP1 and FUS?

We do agree with the reviewer that it can probably be possible to perform FRET experiments (either conventional FRET or FRET by FLIM) and calculate FRET efficiency to support our dc-FCCS data. However, as shown in Fig.4g, the fluorescence lifetime of G3BP1 decreases in the SGs even in the case of WT cells, where FUS is not present in the SGs. The decrease in the fluorescence lifetime of G3BP1 only suggests that there is already a non-radiative exchange of energy (like in homo-FRET) within G3BP1 protein. We think that FRET experiments between G3BP1-gfp and FUS-rfp would not be as definitive and explanatory as the dc-FCCS experiments.

9) Page 11: For the analysis of FLISM data, the authors use an ImageJ plugin and apply a single exponential decay model. Usually, for improving the fitting, it is necessary to use the instrument response function (IRF). Do the authors have acquired an experimental IRF? It could be interesting to see this IRF because it could be dependent on the SPAD array pixel and on the emission wavelength.

We measured the IRF of the full systems and reported it in here. We measured the IRF for both diode lasers used in this work and for the different pixels of the SPAD array. We do not used scattering but a rigorous method, based on the quenching of fluorophores, i.e., based on a fast fluorescence emission. Specifically, we used samples of freely diffusing ATTO495 and Abberior

LIVE 560 fluorophores. Two samples were made for each fluorophore: a solution saturated with potassium iodide to stimulate fluorescence quenching as well as without quencher.

The results are depicted in the figure below.

Figure: Top row: start-stop histograms for quenched (left) and unquenched (right) ATTO495 for different pixels of the SPAD detector. For visualization purposes, the curves for the different pixels are horizontally (and vertically) shifted. Intrinsic delays for the different pixels have been corrected. Bottom row: same plots for Abberior 560.

From the quenched data, we got a FWHM of the IRF of $(28 \pm 2) \cdot 10$ ps for ATTO495 and $(19 \pm 1) \cdot 10$ ps for Abberior560 (average and standard deviation over the five plotted pixels). This is about one order of magnitude lower than the fluorescence lifetimes measured for the samples in the manuscript. Hence, the effect of the IRF on the retrieved lifetime is minimal and can be neglected. In fact, the lifetimes measured for the unquenched samples were (1.06 ± 0.02) ns and (2.23 ± 0.04) ns for ATTO495 and Abberior 560, respectively (average and standard deviation over the five plotted pixels, no deconvolution applied). These values are in very good agreement with the literature values of 1.0 ns (ATTO495) and 2.3 ns (Abberior 560). In addition, we are mainly interested in lifetime differences and changes in the manuscript, rather than the absolute values. For these reasons, we decided not to apply deconvolution or reconvolution as these algorithms may introduce unwanted artifacts. We are aware that the real absolute lifetime values may slightly differ from the ones reported here, but this is due to a small systematic error that does not change anything in the conclusion.

Recognizing the critical importance of conducting rigorous analyses of time-resolved measurements, our research group has initiated a project dedicated to developing an open-source software framework for the analysis of time-resolved datasets, particularly within the context of fluorescence lifetime studies.

Furthermore, we would like to draw attention to the extensive characterization of the precision of the SPAD array detector employed in this work, as well as the BrightEyes-TTM platform. These characterizations were previously detailed in two separate publications: (Buttafava, M. et al. in *Optica* 7:755, 2020) and (Rossetta, A. et al., *Nat. Comm.*, 13(1): 7406, 2022), respectively. The primary distinction in the IRFs among various elements of the SPAD array, within the context of our fluorescence lifetime experiments, lies in the temporal delay. This discrepancy arises due to differences in the internal electronics of the SPAD array detector and the FPGA-based Time-to-Digital Converter (TDC) implementation, which can lead to variations on the order of hundreds of picoseconds. However, it's worth noting that this temporal delay is easily correctable during the subsequent analysis phase. Importantly, we have not observed any dependence of the IRF on photon flux or wavelength.

10) Supp. Material, figure S1: b) the FCS curve in red (monomeric alpha synuclein) is not well fitted with a single component model. Why the authors do not consider a two components model? c) Is it possible to add the diffusion coefficient for monomeric alpha synuclein and also for the first component of LLPS and 10% 1,6-hexanediol.

Fitting model selection is performed according to the Bayesian Information Criterion (BIC). Since BIC associated to single component model (BIC = -2733) is lower than BIC associated to two-component model (BIC = 2727), one-component model has been selected. In any case, given the same BIC, the one-component model is preferred to avoid overfitting by adopting a multi-parameter fitting model.

We apologize if the value of the diffusion coefficient related to monomeric alpha-synuclein was not clearly stated. We reported the value in the main text, page 7 (par. 2.1):

“We characterize the dynamics of monomeric α -syn before the LLPS process. Imaging only shows a uniform solution of fluorescent α -syn. Likewise, svFCS reveals one diffusing component. As expected, α -syn is freely moving in solution with a diffusion coefficient of $D = (130 \pm 7) \mu\text{m}^2/\text{s}$, comparable with previously reported values for monomeric α -syn.”

We also implemented the required changes by adding in figure S1 the diffusion coefficients related to the first diffusing component in LLPS conditions and under 10% 1,6-hexanediol in figure S1:

“Autocorrelation curves of α -syn in the three conditions depicted in (a). Red for α -syn in 20 mM PB, green for α -syn in LLPS buffer, and purple for dissolved α -syn condensates. While α -syn in 20 mM PB can be analyzed with one diffusing component only, α -syn after the LLPS and after the addition of 10% 1,6-hexanediol are analyzed with a two-components FCS model. A faster ($D = (20.0 \pm 1.5) \mu\text{m}^2/\text{s}$ for α -syn LLPS, $D = (18.3 \pm 0.5) \mu\text{m}^2/\text{s}$ for α -syn treated with 10% 1,6-hexanediol) and a slower ($D = (0.5 \pm 0.1) \mu\text{m}^2/\text{s}$ α -syn LLPS, $D = (0.8 \pm 0.2) \mu\text{m}^2/\text{s}$ for α -syn treated with 10% 1,6-hexanediol) diffusing components have been detected.”

11) Supp. Material, figure S6: Why the diffusion coefficients of the YG beads and the red beads which have the same diameter (20 nm) are so different (more than two orders of magnitude)? Normally, the diffusion coefficient should be only related to the size of the bead.

We corrected the legend of Figure S6 and added the values of the diffusion coefficients measured for the two beads. As expected, the value of the diffusion coefficients we obtained is well corresponding with the diameter of the beads. For a better understanding of circular scanning-FCS, we added a brief description of the analytical form of the equation and its parameters in the supplementary info.

Minor issues

1) Page 2, left column: “it is able to reveal also fast-scale (in the range of the measurement time, i.e., tens of seconds)”. I understand the explanation of the authors but the term fast scale may be not well adapted to describe a process of tens of seconds.

We agree with the reviewer that tens of seconds cannot be considered conventionally fast scales. We changed the wording in the manuscript for this specific point. However, it is possibly faster than conventional svFCS on confocal microscopes, where the pinhole must be sequentially adjusted for every different volume.

2) In figure 2 f) and h), please add some details about the pixels used for obtaining the cross-correlation curves. Is it a single element or a sum? Is it possible also to enlarge these 2 graphs because it is hard to visualize clearly the grey curves.

We thank the reviewer for spotting this lack of clarity. The cross-correlation has been calculated between the single-element APD detector and the sum of the inner 3x3 pixels of the SPAD array detector. As reported in the supplementary figure S8, the area of the two detectors is similar. We slightly enlarged the cross-correlation panels.

3) Page 5, left column: please correct the sentence: “Consequently, the cross-correlation also curves have low amplitudes.”

We implemented the required changes.

4) Page 5, left column: please rephrase this sentence: “If, on the other hand, we measured only yellow-green fluorescent beads but with a 50-50 beam-splitter in the detection path and with green detection filters on the single-element detector, we can retrieve both good autocorrelation curves and a cross-correlation curve with high amplitude.”

We implemented the required changes.

5) Page 7, right column: please correct the sentence: “By a simple centroid-based segmentation based on...”.

We implemented the required changes.

6) Page 8, figure 5: “d Time decay histograms for G3BP1 (cyan) and FUS (magenta) in SK-N-BE mutates cells (d)”. I think that it should be (b) instead of (d).

We implemented the required changes.

7) Page 11, left column: “SK-N-BE cell lines stabling expressing...”. Please replace "stabling" by "stably".

We implemented the required changes.

8) Page 11, left column: please check the parenthesis of the sentence: “For the monomeric α -syn in non-LLPS condition, 700 nM Atto488-maleimide α -synC141) was diluted in 20 mM phosphate buffer and 100 mM KCl (Sigma-Aldrich); for the LLPS experiment, Atto488-maleimide α -synC141) and wild-type α -synC141) were mixed with a ratio of 1:20 in the LLPS buffer (20% poly-ethylene glycol 8000 (PEG-8K, P1458, Sigma-Aldrich in 20 mM phosphate buffer.”

We implemented the required changes.

9) Page 11, right column: Is it possible to indicate the value of the beam waist used ? Usually, the calibration of the FCS measurements is performed on a solution of known diffusion coefficient, in order to fit only the amplitude and the focal volume. In this work, the authors fit three parameters: amplitude, diffusion time and focal volume. Why do they use this procedure that is known to lead to more variability?

We added the value of the beam waist retrieved from the circular scanning calibration with fluorescent beads. This measurement is also reported in Supplementary Figure S6. Indeed, for circular-scanning, FCS three parameters instead of two are free during the fitting procedures. It allows for calibration of the detection volume without any a-priori interpretation of the samples. Fluorescent beads are known to be aggregation-prone or Rhodamine-dyes are known not to be always photostable.

10) Supp. Material, page 1, equation S10: The introduction of the factor 0.35 is not explained. Where does it come from?

We are sorry for the confusion. The factor 0.35 is a numerical factor (called gamma factor) dependent on the type of illumination used, one or two-photon (0.076). It describes the shapes of the PSF (3D Gaussian for the confocal and 3D Gaussian-Lorentzian for two-photon excitation). Only the average number of molecules per volume, not the diffusing time, is proportional to this value. As it is only a constant numerical factor, in many applications, this value is simply set to 1. (Lakowicz, J.R. (2006), Principles of Fluorescence Spectroscopy. 3rd Edition, Springer, Berlin. <http://dx.doi.org/10.1007/978-0-387-46312-4>)

For simplicity, we changed the manuscript, setting it to 1.

11) Supp. Material, figure S2: the authors write: “b Dual-color Fluorescence Cross-Correlation Spectroscopy measurement of red fluorescent nanospheres in water.” I am quite confused. Is it red or orange fluorescent beads? and after: “equipped with the orange filter”. Is it red or orange filter?

We thank the reviewer for having spotted this inconsistency. We used two types of fluorescent beads: yellow-green (YG carboxylate fluoSpheres, exc./em. 505/515 nm, Invitrogen) and red (Red carboxylate fluoSpheres, exc./em. 580/605 nm, Invitrogen), with the green or red emission band filter respectively. We change the text accordingly.

12) Supp. Material, figure S5: Is it possible to show the distribution of the confinement strength and the apparent diffusion coefficient, in order to compare more easily with the results of figure S3 and S4?

We created a new supplementary image (S7), which shows both the distribution of confinement strength and apparent diffusion coefficient for G3BP1 (in the case of SK-N-BE cells with FUS^{P525L}) and their trend over time during the recovery. Regarding the distributions, we decided to sum the data measured during the whole recovery (5 hours). For this reason, the boxplot of both D and S_{conf} during the recovery overlaps with both plots of G3BP1 before and after the stress. While the diffusion coefficients increase over time during the recovery (SGs are dissolving), the confinement strength trend is less obvious, also confirming our measurements on G3BP1 before and after the stress, where there was no clear trend of S_{conf} .

13) Supp. Material, figure S6: “The detecting volume for this measurement is”. Please finish this sentence.

We implemented the required change.

Reviewer #2 (Remarks to the Author):

In this manuscript, the authors propose a multi-modal microscopy framework that, first, uses single photon avalanche diode (SPAD) arrays as an alternative method for implementing spot-variation Fluorescence Correlation Spectroscopy (svFCS). Second, they also add a single SPAD detector in conjunction with the SPAD array to the microscope setup in order to perform dual-color Fluorescence Cross-Correlation Spectroscopy (dcFCCS) experiments. This framework allows usage of a single dataset obtained using the whole platform for performing many different types of quantitative analyses to extract dynamical information across spatial and temporal scales. The authors demonstrate in detail the applicability of their framework by observing stress-granule (SG) formation and protein dynamics under their microscope and then analyzing the obtained datasets.

Existing svFCS techniques modify the size of the microscope's back-aperture to change the observation (or light integration) volume while collecting photons using a single detector. On the contrary, the proposed framework uses a detector array (5×5) to collect the photons. The integration is then done as a post-processing step where the observation volume or equivalently the number of detectors in the array to be used for integration is a user choice.

The proposed framework seems to have clear advantage over the existing svFCS methods in terms of temporal resolution, as dynamical information is being collected over different observation volumes simultaneously unlike other methods where information over different volumes is collected sequentially.

The manuscript is well-written and the scientific contribution is significant given that the individual techniques presented here were developed by the same authors and published earlier in Nature Communications (13, Article number: 7406 (2022)) and Nature Light (10, Article number: 31 (2021)). I recommend the publication of this work in Nature Communications with the following comments addressed.

We thank the reviewer's comments on the manuscript. To match also the request from Review #1 and clarify the novelty of this work and its aim also in respect to our previous works, we rewrote the abstract and we change part of the introduction.

We addressed all the points raised by the referee. Changes are indicated in red in the manuscript. A detailed point-to-point answer is written below.

Comments:

1) A general comment for the authors would be that, given the wider audience of Nature Communications which includes both theorists and experimentalists, it would improve the readability of the paper if they define technical phrases clearly or use more descriptive phrases. For example: In the second paragraph on page 1, I am not sure what "sub-resolution" is referring to. Is it sub-diffraction limit or sub-camera pixel size? Similarly, I am not sure what the phrase "Content-Enriched" in the manuscript title is referring to. I have typically seen such phrases being used for describing deep-learning based computational microscopy algorithms that take into account prior information about a

sample. For instance: see this paper “<https://www.nature.com/articles/s41592-018-0216-7>”; published in Nature Methods about a technique called Content-aware image restoration (CARE). Therefore, it would be useful to either clarify such phrasing in the manuscript or come up with a different title to avoid any confusion.

We thank the reviewer for the comments. We reformulated the technical phrases throughout the whole manuscript, and we changed the title.

2) In panel c of Fig. 2, I suspect the x-axis variable should be ω_0^2 as ω_0 should have units of length scale according to the FCS formulation provided in the Supplementary.

We implemented the required change.

3) It would also be useful to explain in the manuscript why a linear relationship is expected in the τ_D vs ω_0^2 plots in Figures 2 & 3, especially when biomolecules being observed are not freely diffusing. From the FCS theory presented in the Supplementary, I can understand that diffusion time is directly related to the observation beam parameter ω_0^2 but it improves readability if it is explained in the main manuscript as well. Naively, one may think that, for heterogeneous samples, behavior may be different at large length scales as compared to shorter length scales such that a linear relationship is not followed. In fact, the authors define the confinement strength S_{conf} to parametrize this exact phenomenon.

We added a short explanation of the diffusion law $\tau_D(\omega_0^2)$ in the supplementary information file, together with the FCS and the circular scanning-FCS theory. We also add this part in the main text, hoping to clarify the diffusion law:

“The diffusion law $\tau_D(\omega_0^2)$ allows distinguishing different modalities of diffusion. In fact, the linear regression of the diffusion times at different observation volumes is described by the function: $\tau_D = \omega_0^2/4D + t_0$. The slope is related to the diffusion coefficient, while the intercept t_0 intuitively describes the time deviation compared to the expected time if the molecule’s movement were solely governed by Brownian motion. This deviation is influenced by the sample environment. In the case of pre Brownian motion $t_0=0$. However, for hopping diffusion t_0 is greater than zero, and for diffusion through a meshwork, t_0 is less than zero.”

4) The definition of confinement strength S_{conf} is not clear to me. Which of the two terms in the ratio form the numerator and the denominator?

The confinement strength here represents the ratio between the diffusion coefficient obtained from the smallest observation area and the one from the biggest observation area. In this case, $S_{\text{conf}} = D_0/D_{5 \times 5}$.

5) For the dsFCCS modality, I do not understand why the authors are using a combination of a single-point SPAD and a 5×5 SPAD array instead of two 5×5 SPAD arrays to obtain equal amount of spatio-temporal information for the two colors.

Indeed, the use of two SPAD array detectors (5x5 or even bigger, like 7x7) could provide more information. With a single measurement, the cross-correlation could be performed and svFCS and lifetime for both species could be performed. However, adding an additional SPAD array detector increases substantially the complexity of the system, both from the hardware and the software point of view. Moreover, the data stream and the data handling would be difficult to handle.

As we hope that our platform will be introduced in many life-science labs to study dynamics, we think that two SPAD array detectors would increase the complexity and reduce the applicability of our platform.

Reviewer #3 Comments:

The authors thank the reviewer for pointing out some key parameters of SPAD (array) detectors, most of which were not described in detail in the previous version of this manuscript but in the cited article Slenders *et al.*, *Biophys. Rep.* **1**, 2021. In the revised manuscript, the authors added a table to the SI, Table S1, indicating the detector specifications. Changes in the manuscript are marked in red. In addition, the authors answer below in more detail to each of the reviewer's questions.

1) Dead time is an important factor for SPAD or SPAD array, which can distortion the photon counting histogram, especially when imaging with high flux dynamic range. What's the parameters of these two used SPAD detectors? and how the author deal with it in the experiments.

Indeed, dead time or hold-off time is one of the key parameters of a SPAD detector. The SPAD array detector used in this work has a tunable dead time between 20 ns and 100 ns. Choosing a lower dead time increases the dynamic range but at the cost of a higher afterpulsing probability. In the context of FCS/FFS, the photon count rate (PCR) is typically very low, in the order of 10-50 kHz, see e.g. Fig. 2 (e, g). In addition, every pixel of the array detector is an independent SPAD, meaning that if a photon hits one pixel, the other pixels remain active. And since the probability of multiple photons arriving at the very same SPAD pixel within the hold-off time is very low, one can safely choose a high dead time without significantly distorting the TCSPC histograms or introducing other artifacts. In fact, in this work, all measurements with the SPAD array detector were done with 100 ns dead time.

The single-element SPAD detector (PD-050-CTC-FC) has a dead time of 77 ns, which leads to a theoretical dynamic range of more than 12 MHz. Thus, also for this detector, typical FCS measurements have a PCR that is orders of magnitude lower than the saturation rate.

2) Fill factor is also an important factor for SPAD array, which affects the collection efficiency of each photons or the lateral resolution of imaging obtained by SPAD array directly. Please give some information about this, such as the pixel pitch, the size of the active area of SPAD pixel in the array

The pixels of the array detector are squares with a size of 50 μm and the pixel pitch is 75 μm . The resulting fill factor is 51%. We added an extra table into the supplementary information of the manuscript to resume all these characteristics.

Notably, the fill factor can be increased by installing a microlens array. We started to use a new SPAD array detector which integrates this microlens, providing an effective fill factor of 80% (see e.g. Zunino *et al.*, *Inverse Problems*, **39**, 2023) but, in this manuscript, all measurements were done with a SPAD array detector without microlenses.

3) The dark count rate of the SPAD array detector should also be described, which is an important noise source for the imaging system. In particular, usually the dark counts of each pixel in the SPAD array cannot be completely consistent, and even hot pixels with a very high dark count rate may appear.

Indeed, each pixel of the array detector has its own characteristics in terms of dark count rate (DCR), afterpulsing etc. It is true that a high DCR can be the main noise source of the (FFS)

experiment and hinder the analysis, as the authors showed in Slenders *et al*, *Biophys. Rep.* 1, 2021. However, in the experiments performed here, none of the pixels of the detector had to be excluded from the analysis or required special treatment. The reason is twofold: on the one hand, the detector was thermo-electrically cooled to a temperature of -15 °C. Cooling reduces the DCR by more than one order of magnitude compared to room temperature. As a result, most pixels have a DCR around or below 100 Hz. Six pixels have a DCR around or above 1 kHz. Pixel 2 (first row, second column) has the highest DCR, equal to 1.7 kHz. On the other hand, the two 'hottest' pixels are in the outer ring of the detector, and their signal only contributes to the 'sum5x5' signal, which is typically in the order of 10s of kHz in total. Thus, the signal-to-noise ratio is always high enough to perform FFS/FLFS.

4) P.9 : 'We additionally integrated a single-element SPAD detector (\$PD-050-CTC-FC, Micro Photon Devices, Italy) to register simultaneously the fluorescence light within two different spectral windows to perform dual-color experiments. Why use two different SPAD to perform dual-color experiments? One SPAD array, and one single-element. What is the advantage over a two-color system built on two unit SPAD detector? or a two-color system built on two SPAD array?'

Indeed, the use of two SPAD array detectors could provide more information, as with a single measurement not only the cross-correlation could be performed but also svFCS and lifetime for both species. However, adding an additional SPAD array detector increases substantially the complexity of the system, both from the hardware and the software point of view. E.g., the data stream and the data handling would become difficult to handle.

As we think that these methods will be spread into life-science labs to study dynamics, we think that two SPAD array detectors would increase the complexity too much.

5) P.10: 'The time tagging module also replicates the input signals of the 3x3 central elements of the SPAD array detector and sends them to the control unit of the microscope.' Is it the SPAD array is used as a single element detector with its 3x3 central elements as one detector element? Is there any difference between the usage of SPAD array in this manuscript and that in Ref [1,2]? which use SPAD array to realize super-resolution microscopy imaging. [1] Tenne R, Rossman U, Rephael B, et al. Super-resolution enhancement by quantum image scanning microscopy[J]. *Nature Photonics*, 2019, 13(2): 116-122. [2] Lubin G, Tenne R, Antolovic I M, et al. Quantum correlation measurement with single photon avalanche diode arrays[J]. *Optics Express*, 2019, 27(23): 32863-32882.

The SPAD array detector can be connected to only one acquisition platform at a time. Thus, when using the BrightEyes-TTM, the detector is not connected to the control unit of the microscope. In order to show the scanned images in real-time, e.g. while scouting a position in the sample, or check that the FFS measurement is working as expected, the BrightEyes-TTM sends a copy of the summed signal from the central 3x3 elements to the control unit of the microscope. This control unit cannot analyze time-tagging information but simply counts photons in time bins and plots intensity information. After the measurement, this information is not used anymore.

We changed the text to make this clearer (section 3.1, last paragraph):

“The time-tagging module also replicates the input signals of the 3x3 central elements of the SPAD array detector and sends them to the control unit of the microscope for real-time visualization of the intensity signal.”

Regarding the comparison with the cited literature, there are indeed some differences. The array detector in [1] does not consist of an array of light-sensitive pixels but instead is made of a fiber bundle that routes the light into 14 individual SPADs. This is a common alternative for ‘true’ on-chip SPAD array detectors with the disadvantage that it is infeasible for upscaling (i.e. when a higher number of pixels is needed). The on-chip detector in [2] is more similar to the one used in this work but with a different number of pixels, arranged in a different pattern, a slightly different technology (CMOS vs. BCD), and commercialized by a different company.

6) Page 10: 'the BrightEyes-MCS software allows recording the fluorescence signal from each SPAD with a sampling (temporal bin) down to 500 ns. Is 500 ns the lowest integral time for intensity based measurement ?

The photon count data is transferred from the FPGA to the PC via a USB 2 connection. Therefore, the software imposes a lower limit of 500 ns on the bin time to prevent potential data loss. This bin time is neither a limit of the detector nor the PC. With a different FPGA with e.g. a PCIe connection, the bin time can be lowered by more than a factor of 5. However, an integration time of 500 ns for the intensity-based measurements is enough to capture dynamics for most biological applications, which usually happen on the time scales of ms – s. If a lower integration time is needed, the fluorescence signal can be acquired in the time-tagging modality (with about 30 picoseconds resolution) and binned to the wanted integration value.

Reviewer #4 (Remarks to the Author):

Paper by Perego et al applies their recently developed FLIM, FCS-based technique (published previously; BrightEyes-TTM) using the power of multi-array SPAD (single-photon avalanche diode) detector to study LLPS formation and dynamics of stress granules(SG)-linked proteins G3BP1 and FUS in cells. Specifically, they were able to determine differences in diffusion coefficients of SG-linked proteins in stressed (with 0.5mM arsenite) and not stressed cells, as well as mutant FUS P525L, and differences in spatial dynamics behavior by invoking the Sconf (confinement) parameter. Overall, the use of new combined technologies is good, it is not clear what new insights was gained with the new technique that was not known previously.

We appreciate the reviewer's comments and their concise summary. As noted by the reviewer, this manuscript builds upon our prior publication in Nat. Comm. (Rossetta, A. et al., Nat. Comm., 13(1): 7406, 2022), where we introduced the BrightEyes-TTM technology. However, that earlier work only focused on the technical aspects of the time-tagging module and provided proof-of-principle validations. The main objective of our present work is to introduce a comprehensive framework that capitalizes on the synergistic capabilities of the asynchronous read-out SPAD array detector and the BrightEyes-TTM multi-channel time-tagging module. This integration empowers us to delve into the investigation of biomolecular processes within living cells, offering an unprecedented level of details across various spatiotemporal scales.

In response to the reviewer's insightful feedback, we have taken the opportunity to revise both the abstract and sections of the introduction. This has allowed us to emphasize the novel aspect of this work and to provide clarity regarding its relationship to our earlier publications. Moreover, we wish to underscore the effectiveness of our framework in elucidating biomolecular condensate processes. In line with this objective, we have incorporated a series of new experiments into the manuscript, which align with the reviewer's suggestions and further demonstrate the power of our comprehensive framework.

Finally, During the revision of this manuscript, we are pleased to report that two additional papers, utilizing this framework to investigate stress granule mechanisms and roles, have been submitted to prestigious journals (Di Timoteo et al., "M⁶A reduction relieves FUS-associated ALS granules", Mariani et al. "ALS-associated FUS mutation reshapes the RNA and protein composition and dynamic of Stress Granules", biorXiv, 2023.09. 11.557245)

We have addressed all the points raised by the referee, and the manuscript has been updated accordingly. Changes are highlighted in red within the manuscript. A detailed point-by-point response is provided below.

Comments:

1. It is difficult to assess the advantage of the multi-SPAD detector vs conventional single SPAD or other fluorescence methods because the conclusions such as slower movements of stressed proteins can be obtained with conventional FLIM, FCS or FRAP technique. Figures or data that shows critical technical advantages are needed, e.g. temporal resolution advantages; i.e., what fast timescales was captured that couldn't be captured previously; spatial resolution inaccessible previously,etc.

We are sorry that we did not manage to directly convey our message. We re-wrote partially the manuscript to better highlight these figures of merit.

2. Where are the FCS spots in the cells? Is there correlation between the intensities of the spots, or the concentration vs the diffusion coefficients or Sconf? Also, it is not clear how significant is the cluster differences (Fig. 3c,d) between the stress and the not stressed, and mutants? Or is it just heterogeneity of the system.

Prior to each measurement, we acquire an image of the field-of-view typically encompassing several cells spanning approximately 50-70 μm . Care is taken to ensure the chosen focal plane is appropriate, and multiple points within the cells are selected randomly for single-point measurements. In general, both the intensity values and the concentration tend to be higher inside stress granules (SGs) compared to before the arsenite treatment. Therefore, there is a degree of correlation between the diffusion coefficients and the intensity, as the condensation process leads to higher concentrations of the labeled molecules within SGs. However, it's important to note that there isn't a clear-cut correlation within individual clusters. We believe that the spread of the cluster is a reflection of the heterogeneity of the system.

3. The nano-environment could be a big advantage for the technique that could have been more explored better to provide more substantial insights on the formation of the stress granules, for examples, there is literature report that SGs have a dynamic outer shell and a more solid core.

Can they provide this temporal and spatial resolution?

If they add RNase to remove RNA in SGs, can they see solidification, changes in the spatio-temporal dynamics and nano-environment. They have also explored only heterotypic interaction between G3BP and FUS, how about the homotypic FUS, FUS cross correlation, which is relevant for irreversible aggregations?

We performed some extra experiments to validate both the biological applications and also to provide more insight into the SGs formation.

We did not find a difference in the nano-environment in SGs marked with G3BP1. In particular, we did not find two different nano-environments, in terms of dynamics or lifetimes or different areas of the SG (core/outer layer) for either mutated or wt cells. This is partially expected as we focused on G3BP1, known to be a marker for only the core of SGs (Wheeler *et al.*, 2016) and not of the outer shell.

However, we tried to answer the question by quantitatively characterizing the SGs shape after treating cells with RNaseA, as suggested. We added a new figure (Figure 5), which summarizes the results of the RNaseA experiment, and we change the text accordingly. We show how we can increase the information provided by our platform by segmenting the SGs of the intensity-based images. The segmentation is directed by the lifetime information: as we measured a shorter lifetime in the SGs compared to the rest of the cells, we selected only the pixels with a lower lifetime, automatically segmenting the SG shapes.

We found that SGs containing mutated FUS are prone to have a lower circularity and a more complex shape compared to the WT SGs. The inhomogeneous structure of SGs containing mutated FUS has also recently been confirmed (Shen *et al.*, 2023).

Upon the presence of RNase, we could measure the unpacking of SGs as a decrease in the area of the SGs and a decrease in the circularity. This experiment address a major critic from the review, since the experiment proves that only with a multi-parameter and an integrative platform biomolecular processes can be exhaustively studied.

Our absolute space resolution limit is technically below the diffraction limit (as we can perform in principle FLIM-FRET experiments to probe the nanometric space regimes) and time resolution

below nanoseconds (we could measure hundreds of picoseconds). Our dataset's ultimate quant of information is at the single photon level.

In principle (not shown here because part of a submitted manuscript), our platform can also be equipped for single-particle-tracking, allowing to track particles with a time resolution below the micro-second.

The advantage of the SPAD array detector in combination with a time tagging system is the access to single-photons data-sets which allows for the implementation of several methods, (ranging from picoseconds to hours and from nm to hundreds of micrometers), restricted only by computational memory and signal-to-noise ratio of the tagged molecules. When the signal-to-noise ratio is high, short integration times (in the orders of milliseconds) are needed to have good svFCS and FLCS measurements.

To further convince the reviewer about the effectiveness of our framework, we decided to share some of these preliminary data which are confidential as part of a submitted manuscripts.

Using the same biological system of all manuscript, we used our platform to gain deeper understanding of the physiology of SGs in ALS systems.

(1) The role of chemical inhibition of METTL3 in ALS.

We investigated how the inhibition of the enzyme METTL3 can affect the SGs dynamics. METTL3 is a methyl-transferase able to chemically modify some RNAs and, in this way, regulating their function. Since the dysregulation of METTL3 has been linked to several pathological conditions, the enzyme is a promising therapeutic target. We found that when METTL3 is chemically inhibited, the dynamics of aberrant SGs (formed by mutated FUS protein) are recovering back to the dynamics of SGs in wt conditions. We think that the inhibition of METTL3 could possibly alter the solid-to-liquid transition of aberrant SGs.

The diffusion coefficients and the confinement strength of FUS^{P525L} are increasing upon inhibition of METTL3 compared to the control sample. This observation has a double implication: (i) FUS proteins can move faster inside SGs, indicating a liquefaction of the SG. (ii) FUS proteins are moving also more freely in SGs, indicating a liquid-like behavior in contrast to a more condensed environment.

Our data were also confirmed by sequencing and molecular biology methodologies.

(2) The role of overexpression of mutated FUS in ALS.

We induced the expression of mutated FUS protein (FUS^{P525L}) for 72 hours instead than 24 h to better denote the overexpression of the mutated protein in the pathological condition. ALS is typically a late-onset disease. The mutated protein is produced over long period of times, and SGs form and dissolve several times before transition from liquid-like to solid-like nature.

We found a stronger confinement for FUS protein when induced longer in cells. We tested here also the role of a protein (marked FBX in the figure below) involved in the ubiquitination pathway, affecting the protein turnover and degradation. We find that FBX can affect FUS dynamics and nano-environment (data not shown), increasing the diffusion coefficient of FUS proteins inside SGs. The shape of SGs is also affected, with a higher circularity and smaller area, suggesting a more liquid environment compared to the control case of only mutated FUS.

To strengthen our claims, we conducted an additional experiment as a suggested control. In this experiment, mutated FUS was labeled with two different fluorescent labels, namely FUS-GFP and FUS-RFP. We performed dual-color cross correlation (dcFCCS) measurements to check on the FUS-FUS interaction. DcFCCS has been performed in the same experimental settings as for the measurement of G3BP1-FUS interactions in the manuscript. FUS-rfp has been measured on the single-element detector, while FUS-gfp with the SPAD array detector.

We used SK-N-BE cells with the same inducible mutated cytoplasmic FUS-rfp (FUS^{P525L}) of the manuscript without the expression of G3BP1-eEGFP. We perform a transient transfection to over-express FUS-gfp (FUS^{P525L}). The signal of FUS-gfp in the cytoplasm is comparable with the signal of FUS-rfp (figure below, panel **a**). While we observe a high signal of FUS-gfp also in the nucleus which can be related to an over-expression artifact. We perform the spectroscopy measurements only in cells expressing both proteins, focusing on measurements inside the granules (once the SGs are formed).

Inside SGs, the autocorrelation curves related to FUS-rfp and FUS-gfp are overlapping (figure below, panels **b** and **c**), as well as the cross-correlation curve, suggesting that the movement of the two differently labeled proteins is the same in both cellular conditions. Once the SGs are formed, we do see a decrease in the fluorescence lifetime ((1.40 ± 0.03) ns measured before the stress, (1.23 ± 0.02) ns measured inside the SGs), suggesting a possible interaction between FUS proteins.

However, because we don't think that this measurement is crucial for the message, we would like to communicate in the manuscript we decided to not include it in the main text.

Minor Comments

There are some spelling or grammatical errors, such as 'manufacture' fig. 2 legend, 'sodium- arsenite' shouldn't have a dash.

We appreciate the reviewer's feedback. We have thoroughly reviewed the manuscript to identify and correct any grammatical errors or typos.

REVIEWERS' COMMENTS

Reviewer #1 (Remarks to the Author):

The authors have satisfactorily answered to the majority of my concerns. I still have few minor comments.

- Concerning the point 3) of my previous review, the authors have replied: *"We considered the apparent diffusion coefficient as the one measured considering all surface of the SPAD array detector (Sum 5x5). The signal coming from all the SPAD array detector channels is integrated, and the autocorrelation is calculated on the summed signal."* As reported here, *"The apparent diffusion coefficients, i.e. the ones measured with the whole detector area"*.

This answer does not totally reply to my concern because I still do not understand how a single apparent diffusion coefficient is deduced when the curves are fitted with a two components model. Is it possible to add more details on this?

- Concerning the point 4), the authors have exhaustively reply to my question in their letter but they do not include them in the article. I think that it could be interesting to add few words about the filtering of the data (mention the fact that they suppress the artefacts related to cell movement or big aggregates and that they average the curves).

- Please remove the parenthesis in: *"As a proof-of-principle, we quantified the fluorescence lifetime value for the YG fluorescent beads, obtaining a value of (1.3 ± 0.1) ns."*

- page 3 in supplementary information, some characters are not readable: *"In the case of Brownian motion $t_0 = 0$, in the case of hopping diffusion $t_0 \neq 0$, and in the case of diffusion through a meshwork $t_0 \neq 0$. For smaller and smaller observation area, the intercept converges to 0."*

- Figure S7 (supplementary information): the legend of the plot b) does not correspond. Please correct it.

Reviewer #2 (Remarks to the Author):

The authors have made satisfactory modifications to the manuscript to address previous comments by myself and the other reviewers, as described in their Response to Referees letter. I recommend publication of this manuscript in its current form.

Reviewer #3 (Remarks to the Author):

I thank the authors for the updated manuscript. All of my concerns and questions have been addressed.

Reviewer #4 (Remarks to the Author):

The revised manuscript was very much improved to address the reviewers' comments. The integrated methodology would be useful and applicable to various studies of biomolecular condensates. The proof-of-concept methods have been applied to two unpublished systems which strengthen the application of the integrated technology.

We thank all the reviewers for their comments and for the time spent revising the manuscript.

Reviewer #1

The authors have satisfactorily answered to the majority of my concerns. I still have few minor comments.

We express our gratitude to the reviewer for their valuable input. We are answering here point-by-point to his comments.

- Concerning the point 3) of my previous review, the authors have replied: “We considered the apparent diffusion coefficient as the one measured considering all surface of the SPAD array detector (Sum 5x5). The signal coming from all the SPAD array detector channels is integrated, and the autocorrelation is calculated on the summed signal.” As reported here, “The apparent diffusion coefficients, i.e. the ones measured with the whole detector area”.

This answer does not totally reply to my concern because I still do not understand how a single apparent diffusion coefficient is deduced when the curves are fitted with a two components model. Is it possible to add more details on this?

We are sorry for not being clear. We acknowledge the lack of clarity here. When the curves were fitted with a two component model (sometimes, with G3BP1 in granules was necessary, and we added it in the SI), we used the diffusion times of the component related to the diffusion in SGs to create the diffusion law. In this case, the apparent diffusion coefficient is the one related to the SG component (the slow component of the fit, while the fast component could be associated with the diffusion in the dilute condition) measured on the whole detector area.

- Concerning the point 4), the authors have exhaustively reply to my question in their letter but they do not include them in the article. I think that it could be interesting to add few words about the filtering of the data (mention the fact that they suppress the artefacts related to cell movement or big aggregates and that they average the curves).

We agree with the reviewer and we added a clearer explanation in the methods and in the reporting summary:

“The individual correlation curves were visually inspected, and all curves without artifacts, introduced by cell movements or bleaching, were averaged.”

- Please remove the parenthesis in: “As a proof-of-principle, we quantified the fluorescence lifetime value for the YG fluorescent beads, obtaining a value of (1.3 ± 0.1) ns.”

We acknowledge the reviewer comment, however removing the parenthesis will decrease the clarity in our manuscript as it will be inconsistent with the rest of the text. We used the parenthesis as both the mean and the standard deviation have a unit, and this is ns. We are confident that using parenthesis is the standard convention in live sciences.

- page 3 in supplementary information, some characters are not readable: “In the case of Brownian motion $t_0 = 0$, in the case of hopping diffusion $t_0 \neq 0$, and in the case of diffusion through a meshwork $t_0 \neq 0$. For smaller and smaller observation area, the intercept converges to 0. “

We fixed the text.

- Figure S7 (supplementary information): the legend of the plot b) does not correspond. Please correct it.

are very thankful for spotting the error, we have corrected the legend.

Reviewer #2 (Remarks to the Author):

The authors have made satisfactory modifications to the manuscript to address previous comments by myself and the other reviewers, as described in their Response to Referees letter. I recommend publication of this manuscript in its current form.

We thank the reviewer for his/her positive comments.

Reviewer #3 (Remarks to the Author):

I thank the authors for the updated manuscript. All of my concerns and questions have been addressed.

We thank the reviewer for his/her positive comments.

Reviewer #4 (Remarks to the Author):

The revised manuscript was very much improved to address the reviewers' comments. The integrated methodology would be useful and applicable to various studies of biomolecular condensates. The proof-of-concept methods have been applied to two unpublished systems which strengthen the application of the integrated technology.

We thank the reviewer for his/her positive comments.